# Biochemical characterization of an alkaline and detergent-stable Lipase from *Fusarium annulatum* Bugnicourt strain CBS associated with olive tree dieback

Ahlem Dab[1,2], Ismail Hasnaoui[1,3], Sondes Mechri[2], Fawzi Allala[4], Khelifa Bouacem[4], Alexandre Noiriel[1], Amel Bouanane-Darenfed[4], Ennouamane Saalaoui[3], Abdeslam Asehraou[3], Fanghua Wang[5], Abdelkarim Abousalham[1]*, Bassem Jaouadi[2]*

1 Institut de Chimie et de Biochimie Moléculaires et Supramoléculaires (ICBMS), Université Lyon, Université Lyon 1, UMR 5246 CNRS, Génie Enzymatique, Membranes Biomimétiques et Assemblages Supramoléculaires (GEMBAS), Villeurbanne, France, 2 Laboratoire de Biotechnologie Microbienne et d'Ingénierie des Enzymes (LBMIE), Centre de Biotechnologie de Sfax (CBS), Université de Sfax, Sfax, Tunisia, 3 Faculté des Sciences d'Oujda (FSO), Laboratoire de Bioressources, Biotechnologie, Ethnopharmacologie et Santé (LBBES), Université Mohammed Premier (UMP), Oujda, Morocco, 4 Faculté des Sciences Biologiques (FSB), Laboratoire de Biologie Cellulaire et Moléculaire (LCMB), Equipe de Microbiologie, Université des Sciences et de la Technologie Houari Boumediene (USTHB), El Alia, Bab Ezzouar, Alger, Algeria, 5 School of Food Science and Engineering (SFSE), South China University of Technology (SCUT), Guangzhou, China

* abdelkarim.abousalham@univ-lyon1.fr (AA); bassem.jaouadi@cbs.rnrt.tn (BJ)

**Data Availability Statement:** All relevant data are within the paper and its Supporting Information files.

## Abstract

This work describes a novel extracellular lipolytic carboxylester hydrolase named FAL, with lipase and phospholipase $A_1$ ($PLA_1$) activity, from a newly isolated filamentous fungus *Ascomycota* CBS strain, identified as *Fusarium annulatum* Bunigcourt. FAL was purified to about 62-fold using ammonium sulphate precipitation, Superdex® 200 Increase gel filtration and Q-Sepharose Fast Flow columns, with a total yield of 21%. The specific activity of FAL was found to be 3500 U/mg at pH 9 and 40°C and 5000 U/mg at pH 11 and 45°C, on emulsions of triocanoin and egg yolk phosphatidylcholine, respectively. SDS-PAGE and zymography analysis estimated the molecular weight of FAL to be 33 kDa. FAL was shown to be a $PLA_1$ with a regioselectivity to the *sn*-1 position of surface-coated phospholipids esterified with α-eleostearic acid. FAL is a serine enzyme since its activity on triglycerides and phospholipids was completely inhibited by the lipase inhibitor Orlistat (40 μM). Interestingly, compared to *Fusarium graminearum* lipase (GZEL) and the *Thermomyces lanuginosus* lipase (Lipolase®), this novel fungal (phospho)lipase showed extreme tolerance to the presence of non-polar organic solvents, non-ionic and anionic surfactants, and oxidants, in addition to significant compatibility and stability with some available laundry detergents. The analysis of washing performance showed that it has the capability to efficiently eliminate oil-stains. Overall, FAL could be an ideal choice for application in detergents.

**Funding:** This work was supported by the French Ministry for Europe and Foreign Affairs (MEAE), the French Ministry of Higher Education, Research, and Innovation (MESRI), the Centre National de la Recherche Scientifique (CNRS), and the Tunisia Ministry of Higher Education and Scientific Research (MESRS) within the framework of the Multilateral Project Partenariats Hubert Curien (PHC)-Maghreb 2020 Program (FranMaghZYM 2020-2024, Code Campus France: 43791TM & Code PHC: 20MAG01), and the Algerian-Tunisian R&I Cooperation for the Mixed Laboratories of Scientific Excellence (Hydro-BIOTECH 2021-2025, code LABEX/TN/DZ/21/01) as well as by the Ph.D. Student Fellowship of the Doctoral Institute of Fundamental Sciences of the Sfax University represented by the Sfax Faculty of Sciences, University of Sfax (Code: ED08FSSf01). The funders had no role in study design, data collection and analysis, decision to publish, or preparation of the manuscript.

**Competing interests:** The authors have declared that no competing interests exist.

**Abbreviations:** FAL, lipase from *Fusarium annulatum* Bunigcourt strain CBS; PLA$_1$, phospholipase A$_1$; GZEL, lipase from *Fusarium graminearum*; Lipolase®, lipase from *Thermomyces lanuginosus*; TC2, triacetin; TC4, tributyrin; TC6, trihexanoin; TC8, trioctanoin; OO, olive oil; NaTDC, sodium taurodeoxycholic acid; egg PC, egg phosphatidylcholine; GA, gum Arabic; β-CD, β-cyclodextrin; PMSF, phenylmethanesulfonyl fluoride; DIFP, diisopropyl fluorophosphate; MUFB, 4-methylumbelliferone butyrate; DTNB, 5,5'-dithiobis 2-nitrobenzoic acid; NEM, N-ethylmaleimide; PAO, phenylarsine oxide; *sn*-EOPC, 1-α-eleostearoyl-2-octadecyl-*rac*-glycero-3-phosphocholine; *sn*-OEPC, 1-octadecyl-2-α-eleostearoyl-rac-glycero-3-phosphocholine; EDTA, ethylenediaminetetraacetic acid; PVDF, polyvinylidene fluoride; PDA, potato dextrose agar; SNA, Spezieller Nahrstoffarmer Agar; TGs, triglycerides; TAED, tetraacetylethylenediamine); Na$_2$CMC, sodium carboxymethylcellulose; STPP, sodium tripolyphosphate; ITS, internal transcribed spacer; TEF1-α, translation elongation factor 1-α; RPB2, RNA polymerase II second largest subunit; DMSO; 2-ME, 2-mercaptoethanol; DL-DTT, DL-dithiothreitol.

## Introduction

Enzymes have become crucial in the detergent industry as consumers look for ways to clean various stains with minimal damage to fabrics and the environment, while ensuring efficiency and ease. In fact, a bio-laundry detergent contains 0.4% to 0.8% enzyme by weight, serving to remove a variety of common stains. After proteases and carbohydrases, lipases are the third largest group of enzymes as regards the market value of commercial detergents [1,2]. Approximately 1000 tons of lipase is projected to be added to 13 billion tons of detergent yearly. By 2023, the microbial lipase market is expected to be worth USD 590.2 Million [3]. The use of lipolytic enzymes in household detergents decreases, or even substitutes, synthetic detergents, eliminating any harmful effects and expanding the detergent's power to eradicate stubborn grease stains or oil from clothes without much wear and tear [4,5].

Lipases (triacylglycerol acylhydrolases, EC 3.1.1.3) are enzymes that catalyze the hydrolysis of triacylglycerols of the ester bond, in water-insoluble carboxylic esters, to glycerol and fatty acids at oil/water interfaces [6–9]. They are broadly found in animals, plants, and microorganisms [10–12]. Nowadays, the use of alkaline lipase-based detergents is favoured over conventional synthetic ones. A lipase used in detergent preparation should preferably be active and stable within a wide range of pH and temperature conditions and compatible with several detergent additives, such as surfactants and oxidizing agents [13]. In this framework, several trials have been launched to provide suitable microbial lipases. Lipolase® (Novozymes A/S, Kongens Lyngby, Denmark) was the first commercial lipolytic enzyme to be used in the detergent industry and it was manufactured in 1994 by overexpressing the fungal lipase from *Thermomyces lanuginosus* in *Aspergilus oryzae* [3,14,15]. Several other microbial lipases have been used in detergents, such as Lumafast from *Pseudomonas mendocina* and Lipomax from *Pseudomonas alcaligenes*, which were produced in 1995 by Genencor International, AU-KBC Research Center, Life Sciences, Anna University, Chennai, India [16,17]. Other enzyme-engineered variants of Lipolase® were later introduced by Novozymes A/S, namely Lipolase® Ultra, LipoPrime™, and Lipex®, as new fat-hydrolyzing preparations with great potential in laundry detergents. A considerable number of fungal lipases have also been produced commercially for use in food processing, such as "Amano" an F-AP15 Lipase generated from *Rhizopus oryzea*, Lipase A from *Aspergillus niger*, Lipase AY "Amano" 30 from *Candida rugosa*, and Lipomod™ a lipase isolated from *Penicillium* sp. [18].

A lipase can be used on materials to guarantee easy oil eradication. It works by establishing a fabric-lipase complex on the surface of the clothing that forms a barrier [19–21] which prevents the enzyme from being removed throughout washing and stops the oily substances from depositing on the material. An appropriate lipase for the laundry detergent industries is required to meet certain constraints. The key requisite is its stability in washing conditions with a high alkaline pH and a broad range of temperatures (from low temperatures for synthetic fibers to elevated temperatures for cotton). Hence, alkaline and thermostable lipases have been investigated as potential additives in cleaners in many studies [22,23]. Another constraint is the compatibility of the lipolytic enzyme with the other detergent components in the preparations [24]. Regardless of the considerable progress made in recent times, the demand for the isolation and commercialization of new strains of fungi producing novel lipases, with distinct catalytic characteristics for use in detergent compositions, remains a crucial issue.

Tunisia is considered as the second or third largest producer and exporter of olive oil (OO) in the world, with over 70 million olive trees and a cultivated area of approximately 1.7 million hectares. The *Olea europaea* trees are among the most highly cultivated tree species in Tunisia [25]. However, the constant threat of phytopathogenic fungi to olive cultivation, causing dieback and wilting, has resulted in significant economic losses in Tunisian olive groves [26–29].

Diagnosis of dieback of young olive trees revealed the presence of a complex soil fungus isolated from the impacted plant tissues [30,31]. Fungi find, in the olive tree, the lipids, proteins and vitamins necessary for their growth and, amongst these microorganisms, *Fusarium* spp. can cause infectious diseases in olive trees. Some fungi, like *Verticillium dahliae* the causative agent of verticillosis, are well-known as shared pathogens on the olive trees. However, there has also been an increase in the frequency of *Fusarium* species as pathogens. *Fusarium* is a well-known phytopathogenic genus affecting the olive trees in Tunisia and probably those in many other countries in the Mediterranean basin. These species have been recognised from various regions of the world [30,32,33].

The goal of this study was to identify and characterize the lipolytic activity in *Fusarium* spp. isolated from infected olive plants in the Orchards of Gabes (El Amarat) and Tataouine (Henchir El Ghazal) (South-East Tunisia). As part of an ongoing project to search for novel attractive microbial hydrolases, we have so far isolated a lipase-producing fungus, a member of the *Nectriaceae* family, identified as *Fusarium annulatum* Bugnicourt strain CBS. To the knowledge of the authors, no enzyme research concerning the species *Fusarium annulatum* has yet been reported. Thus, the primary objective of this research is (i) to identify some *Fusarium* spp. isolates obtained from olive plants that have been affected, (ii) to report, for the first time, the identification, purification, and physico-chemical proprieties of a new (phospho) lipase, designated FAL, from a recently discovered *Fusarium annulatum* Bugnicourt strain CBS (iii) to evaluate the compatibility of this new lipase with oxidizing agents, surfactants and commercially available detergents and, finally, (iv) to test the washing efficiency of FAL on oil-stained fabric.

## Materials and methods

### Material

Triacetin (TC2), tributyrin (TC4), trihexanoin (TC6), trioctanoin (TC8) (99%, puriss), olive oil (OO), rhodamine B, benzamidine, sodium taurodeoxycholic acid (NaTDC), egg phosphatidylcholine (egg PC), gum Arabic (GA), β-cyclodextrin (β-CD), 4-methylumbelliferone butyrate (MUFB), 5,5'-dithiobis 2-nitrobenzoic acid (DTNB), yeast extract, N-ethylmaleimide (NEM), phenylarsine oxide (PAO), iodoacetamide, and chemical reagents were from Sigma-Aldrich Chimie (Saint-Quentin-Fallavier, France). Synthetic phospholipids were also esterified with α-eleostearic acid [1-α-eleostearoyl-2-octadecyl-*rac*-glycero-3-phosphocholine (named *sn*-EOPC)] and [1-octadecyl-2-α-eleostearoyl-rac-glycero-3-phosphocholine (named *sn*-OEPC)] at the *sn*-1 and *sn*-2 positions, respectively. So, these substrates contain α-eleostearic acid with strong (UV) absorption properties, as well as a non-UV-absorbing alkyl chain with a non-hydrolyzable ether linkage at the other *sn* positions to prevent acyl chain migration during the lipolysis reaction. Casein peptone was from Carl Roth (Lauterbourg, France). ethylenediaminetetraacetic acid (EDTA), polyvinylidene fluoride (PVDF) transfer membrane and molecular mass marker proteins were from Euromedex (Souffelweyersheim, France). Orlistat, a digestive lipase inhibitor, was purchased from Hoffmann-La-Roche Ltd in Basel, Switzerland and Lipolase®, a 1,3-specific *Thermomyces lanuginosus* lipase (TLL) was provided by Novozymes Biopharma DK A/S (Bagsvaerd, Denmark). GZEL was purified as described previously [34]. The Cytiva Lifescience™ Superdex® 200 Increase 10/300 GL column 200 Increase 10/300 GL Prepacked Tricorn™ size exclusion chromatography column (L × I.D. 30 cm × 10 mm, 8.6 µm particle size), the ÄKTA™ prime FPLC™ protein purification system and the Cytiva Life Sciences™ HiTrap™ Q-Sepharose Fast Flow (FF) columns, with a 5 mL volume, were all from GE Healthcare, Bio-Sciences AB (Uppsala, Sweden).

## Sampling and fungal isolation

The olive groves were from two separate geographical governorates, Gabes (El Amarat, GSP coordinates: 34˚33' N, 10˚07' E) and Tataouine (Henchir El Ghazal, GPS coordinates: 33˚40' 12N, 10˚38' E), in the South-East of Tunisia, with different management practices. Both regions are characterized by an arid climate. Fungal isolates were taken from soil-borne fungi and from the major species of olive tree, *Olea europaea* cv. Chemlali. A total of 67 fungal isolates were collected and identified corresponding to their phenotypic and biochemical proprieties, as well as their biological activities. The fungi were isolated using two separate methods. The first method consisted of transferring the superficial mycelium developing on the epidermis of the olive root into Petri dishes containing potato-dextrose-agar (PDA) supplemented with streptomycin sulphate (0.1 g/L). Infected root tissues were made germ-free with NaCl (1%) for 5 min, then treated with ethanol (96%) and, finally, washed with sterilized bi-distilled water. Aseptic fragments of approximately 0.6 cm × 0.6 cm were removed from the tissues and placed on PDA. Culture plates were incubated in the dark and maintained at 26˚C for one week. Hyphae tips from the colonies were subsequently moved to fresh PDA plates in order to get pure cultures. These pure cultures were obtained after two transfers to fresh agar plates and they were verified using microscopic examination to confirm purity and strain identity. Purified fungal isolates were cultured on PDA medium and conserved at 4˚C.

## Screening of lipase-producing microorganisms and culture conditions

Screening for lipase-producing microorganisms was monitored using PDA plate assays supplemented with OO at 1% (*v/v*) and rhodamine B at 0.01% (*w/v*). Culture plates were stored at 30˚C, for 5 days, and any colony displaying orange halos over the mycelium upon UV irradiation was considered as a putative lipase producer [35].

To achieve elevated lipase productivity, the fungal strain CBS was precultured with continuous shaking, at 160 rpm for 24 h and at 25˚C, in 250 mL shaking flasks containing 50 mL of medium A. The medium comprised (in g/L) casein peptone (15), yeast extract (5), $KH_2PO_4$ (1.75) and $MgSO_4$ (0.5), at pH 6. Two milliliters of *Fusarim annulatum* pre-culture was taken to inoculate 2-L shaking flasks containing 500 mL of medium A with OO at 1% (*v/v*), as an inducer for the expression of lipase and as a carbon source, or glucose (15 g/L) as the only source of carbon. The fungus was grown aerobically for 5 days, at 25˚C, on a rotary shaker at 150 rpm. Samples were taken daily for 8 days.

## Morphology and molecular characterization of CBS isolated fungi

The mycelium fragments (6 cm × 6 mm) of the isolate CBS-09 were moved to the centre of the 90 mm diameter plates. Four replica PDA plates were then incubated in the dark, at 26˚C, to evaluate colony growth. Colonies were cultured on Spezieller Nahrstoffarmer Agar (SNA) [36], with a 12 h light/dark incubation to assess microscopic features. The growth of the mycelium was measured by determining the perpendicular and horizontal diameters of the colonies using a caliper. The average of these dimensions was registered after 4 days of PDA-incubation or 6 days on SNA. The colony color was assessed after 14 days of growth. Samples were prepared for microscopic examination after ten days of incubation and observations were made on both media [37]. The criteria applied were those established by Bugnicourt in 1952, as well as by Nelson et al. [38] and Yilmaz et al. [39]. These criteria took into account various factors, such as the presence of conidiophores and conidia on aerial mycelium, conidiogenous cells, and the production of sporodochia and chlamydospores. The measurements (mean, maximum, and minimum) were recorded for each of the 30 identified structures. Morphological

characteristics were detected using an Olympus BX50F optical microscope and photomicrographs were captured with an Olympus SC20 camera to document the findings.

## Molecular identification and phylogenetic analysis of CBS isolate

Gene sequencing of the internal transcribed spacer (*ITS*) and translation elongation factor 1 α (*TEF1-α*), as well as the RNA polymerase II second largest subunit (*RPB2*), was carried out to identify the genus to which the CBS isolate belonged. Fresh mycelium was used to extract total genomic DNA using the Animal and Fungi DNA Preparation Kit (Jena Bioscience, GmbH, Germany). The identity of the CBS isolate was determined based on the macroscopic and microscopic morphological descriptions using the method reported by Visagie et al. [40]. The CBS isolate was identified using DNA extracted with a Qiagen DNeasy Ultraclean Microbial DNA Isolation Kit and fragments of the *ITS* 1 and 2 regions, comprising the 5.8S rDNA, were analyzed at the Westerdijk Fungal Biodiversity Institute in Utrecht, Netherlands. The molecular identification of the CBS isolate was performed at the Westerdijk Fungal Biodiversity Institute (CBS, Utrecht, Netherlands) using the following procedure: DNA was extracted using a Qiagen DNeasy Ultraclean™ Microbial DNA Isolation Kit. Fragments of the *ITS* 1 and 2 regions, including the 5.8S rDNA (*ITS*), a portion of the *TEF1-α* and fragments of the RPB2, were amplified and sequenced. The primers used were *ITS*: LS266 (GCATTCCCAAACAACTCGACTC) and V9G (TTACGTCCC TGCCCTTTGTA); *EF1α*: EF1-728F (CATCGAGAAGTTCGAGAAGG) and EF2 (GGARGTACCA GTSATCATGTT); *RPB2*: RPB2-5F2 (GGGGWGAYCAGAAGAAGGC) and RPB2-7CR (CCCA TRGCTTGYTTRCCCAT). PCR amplification, DNA electrophoresis and purification, restriction, ligation and transformation, as well as DNA sequencing and phylogenetic analysis, were all achieved according to the methods recently defined by the authors [1,2].

## Lipolytic activity measurements

Lipolytic activity was potentiometrically assayed by automatically titrating the released free fatty acids from mechanically stirred triglycerides (TGs) or egg PC emulsions, using NaOH 0.1 N solution and a pH-STAT device (Metrohm 902 Titrando, Herisau, Switzerland). The pH was adjusted to 9 and the FAL solution was added at zero time after recording the background level for 5 min to10 min.

   TG assay: Each single assay was achieved in a thermostated (40°C) vessel comprising 0.5 mL TGs substrates (TC4 or TC8) and 9.5 mL of Tris–HCl buffer (2.5 mM at pH 9), supplemented with 2 mM $CaCl_2$ and 2 mM NaTDC. Long-chain TGs (OO) had first to be pre-emulsified with GA, as previously described [41]. Briefly, the OO emulsion was prepared by combining (3 s × 30 s in a Waring blender) 10 mL of OO in 90 mL of 10% GA. Next, 5 mL of this emulsion were combined with 20 mL of 2.5 mM Tris–HCl (pH 9) plus 2 mM $CaCl_2$, in the pH-STAT vessel.

   Egg PC assay: the assay was accomplished as previously reported [42]. 4 g of reagent-grade PC (egg yolk) was mixed in 100 mL of 4 mM $CaCl_2$ and filtered with cheesecloth. 5 mL of the resulting substrate solution was added to 10 mL of sodium deoxycholate (20 mM) and mechanically emulsified.

   When FAL was assayed without $CaCl_2$, EDTA—the common chelating agent for divalent ions, was added to the assay system. Under the aforementioned test conditions, one international lipase or phospholipase unit (U) corresponds to the release of 1 μmol of fatty acid per min. Specific activities were expressed in U per milligram of protein.

## Purification of FAL

The culture medium (500 mL), retained after 5 days of culture of the CBS fungal strain when the lipase activity is at a maximum, was initially filtered through a Whatman grade No. 1 filter

paper (110 mm size) to remove the mycelium. The filtrate was then centrifuged for 20 min at $7500 \times g$ to further clarify the supernatant. Following centrifugation, the supernatant was filtered again using a Whatman grade No. 1 filter paper (110 mm size) to ensure complete removal of any remaining mycelium and debris. The supernatant, containing extracellular lipase and constituting the crude enzymatic extract, was precipitated and concentrated using ammonium sulphate, at a concentration of 80% saturation, under gentle stirring (4°C). Protein precipitate was recovered by centrifugation at $8000 \times g$ for 30 min, at 4°C, and resuspended in 5 mL of Tris-HCl buffer (20 mM at pH 9) added with 20 mM NaCl. The obtained sample (5 mL) was loaded onto a Cytiva Lifescience™ Superdex® 200 Increase 10/300 GL column 200 Increase 10/300 GL Prepacked Tricorn™ gel filtration chromatography Column (L × I.D. 30 cm × 10 mm, 8.6 μm particle size), which had already been equilibrated with buffer A. Proteins were then eluted with the identical buffer, at a flow rate of 1 mL/min, and fractions of 1 mL were collected. Lipase activity was confirmed, as explained above, and the elution outline of the proteins was measured at 280 nm. Fractions containing lipase activity were assembled, extensively dialyzed against 20 mM Tris-HCl buffer pH 9 (buffer A) and then loaded onto a HiTrap™ Q-Sepharose FF (Cytiva, Sigma-Aldrich) column (5 mL) which had been pre-equilibrated with buffer A. The column was washed with at least 5 column volumes of the buffer A and adsorbed proteins were then eluted with a linear NaCl gradient (0–0.5 M NaCl in buffer A), at a flow rate of 1 mL/min. Chromatographic fractions of 1 mL, comprising FAL activity, were pooled and used for lipase activity characterization.

## Protein determination, electrophoresis, and zymography

The protein content was determined by the Bradford method [43], using Bio-Rad Protein Assay Dye. SDS-PAGE [12% in the presence of 2-mercaptoethanol (2-ME)], according to the Laemmli technique, was used to examine the active chromatographic fractions of lipase and phospholipase [44]. Next, the protein bands were visualized with Coomassie Brilliant Blue G-250 stain from Bio-Rad Laboratories, Inc., (Hercules, CA, USA). The molecular weight scale used for the experiment was the Protein ladder Plus prestained (#06P-0211, Euromedex). Different fractions were tested by zymography using MUFB as the substrate. Enzyme extracts were neither boiled nor treated with 2-ME before loading onto the gel. After protein migration, gels were soaked at 25±2°C in Triton X-100 (2.5%) for 30 min, washed in 50 mM phosphate buffer (pH 8), and covered with MUFB solution (100 μM) in the same buffer [45]. Lipase activity bands became viewable in a quick time after exposure to UV-light.

## NH$_2$-terminal sequence analysis

After purification, the FAL sample was subjected to SDS-PAGE analysis and transferred to a PVDF membrane using the Bio-Rad Trans-Blot® electrophoretic transfer cell, following the manufacturer's guidelines. The membrane was double washed with bi-distilled water and then stained with Ponceau red. The amino acid sequences of the FAL protein in the stained bands were determined on the PPSQ-31B model protein sequencer (Shimadzu Co., Kyoto, Japan) through automated Edman degradation.

## Biochemical characterization of the purified FAL

**Effect of pH and temperature on FAL activity and stability.** To investigate the optimal pH of FAL, the activity was monitored at different pH values between 4 and 11, at 40°C and 45°C. The pH stability of FAL was measured by pre-incubating the enzyme at 25±2°C during 1 h in the following buffers: 50 mM sodium acetate (pH 4–6), 50 mM potassium phosphate (pH 6–8), 50 mM Tris-HCl (pH 7–9), or 50 mM glycine-NaOH (pH 8–12).

The FAL thermoactivity was assessed at varying temperatures, from 25˚C to 55˚C, using a temperature-controlled cell holder. For the FAL thermostability, purified enzyme samples were incubated at four different temperatures (30˚C, 37˚C, 40˚C, and 45˚C) for varying times (5 min, 15 min, 30 min, 60 min, and 90 min). After the incubation period, the remaining lipase and phospholipase activities were determined using the standard assay conditions after centrifugation. The enzyme that was not heated and was kept at 25±2˚C was taken as 100% (control).

**Effect of Orlistat on FAL activity.** Orlistat, a potent inhibitor of digestive lipase, is an analogue of lipstatin that was originally isolated from *Streptomyces toxytricini* [46]. Studies have demonstrated that the interaction between the open β-lactone ring of orlistat and the catalytic serine residue of pancreatic lipase forms a stable acyl-enzyme complex with a long half-life, effectively inhibiting the enzyme in a stoichiometric manner. The hydrolysis rate of the TC8 or egg PC by FAL was measured with Orlistat (40 μM, final concentration) at 40˚C and pH 9.

**Effects of inhibitors, reducing agents, chelating reagents, metal ions, and bile salts on FAL activity.** The effect of various inhibitors and reducing agents, such as phenylmethanesulfonyl fluoride (PMSF), diisopropyl fluorophosphate (DIFP), benzamidine, iodoacetamide, DTNB, PAO, NEM, EDTA, EGTA, 2-ME, and DL-dithiothreitol (DL-DTT), in addition to metal ions with a concentration of 2 mM ($Ca^{2+}$, $Fe^{2+}$, $Mn^{2+}$, $Mg^{2+}$, $Ba^{2+}$, $Zn^{2+}$, $Cu^{2+}$, $Co^{2+}$, $Ni^{2+}$, $Hg^{2+}$, and $Cd^{2+}$), on FAL stability was established by pre-incubating the FAP protein with each of these agents or ions at 40˚C for 1 h. Afterwards, lipase tests were performed under standard assay conditions. To evaluate the influence of bile salts on enzyme activity, FAL activity was observed in several NaTDC concentrations [1 mM—18 mM]. The FAL activity was monitored continuously using the pH-STAT technique under standard conditions.

**Substrate specificity of FAL.** The substrate specificity of FAL was evaluated using various TGs substrates with varying chain lengths: TC2, TC4, TC6, TC8, and OO emulsion. The enzymatic activities were measured using a titrimetric method. Additionally, a continuous spectrophotometric assay was performed to examine the regioselectivity of FAL towards phospholipids, using *sn*-EOPC or *sn*-OEPC as the coated substrate [47]. The wells of a microtiter plate were covered with artificial TGs and phospholipids, as previously reported [47]. The microtiter plate wells that were coated with substrate were thoroughly washed with 10 mM Tris-HCl buffer at pH 8 added with CaCl2 (6 mM), NaCl (150 mM), EDTA (1 mM), and β-CD (2.5 mM). They were then allowed to equilibrate in the reaction buffer (200 μL) at 40˚C for 10 min. FAL (0.65 μg) was then added to the microtiter plate wells and the absorbance was monitored at 272 nm using a microplate reader spectrophotometer (Tecan Infinite® M200 Pro, Tecan Life Sciences Trading AG, Switzerland) at regular 1-min intervals. The microtiter plate was shaken during 5 s previous each reading. Next, the absorbance values were compared against the control, which was the buffer alone. The FAL specific activity was assessed by calculating the change per min in absorbance, employing α-eleostearic acid and a molar extinction coefficient of 5320 $M^{-1}$ $cm^{-1}$ [47].

## Performance evaluation of the purified FAL compared with GZEL and Lipolase®

**Effect of various organic solvents on FAL stability and tolerance.** The tolerance and stability of FAL in organic solvents was assessed by incubating the lipase preparation with numerous organic solvents at 25% (*v/v*), with different Log P values, evaluated by exposing the enzyme to each solvent for 24 h at 25±2˚C with constant stirring at 200 rpm. The partition coefficient, Log P, values are a quantitative measure of solvent polarity, which can help assess

enzyme stability in organic solvents. Samples of the mixture were taken at regular intervals to measure any remaining FAL activity, with the reaction mixture without any additive considered as the control (100%).

**Effect of some detergent additives on lipase stability.** The suitability of FAL as a biodetergent ingredient was measured by assessing its stability with some bleaching agents e.g., $H_2O_2$ and $NaBO_3$, surfactants, zeolite, SDS (linearalkylbenzene sulfonate) and LAS (sulfobetaine), nonionic surfactants (Triton X-100 and Tween-20, -40, -60, and -80), anti-redeposition agents [tetraacetylethylenediamine (TAED), $Na_2CO_3$, sodium tripolyphosphate (STPP), and $Na_2CMC$]. Other detergent components were also assayed as previously reported by the authors [1]. The remaining lipase activity was determined at 40°C and pH 9. The control FAL activity in the absence of any additive was taken as 100%.

**Compatibility and stability of FAL with laundry detergents.** Both the compatibility and stability of FAL, GZEL, and Lipolase® with various laundry detergents have been studied, as recently described by the authors [1] and elsewhere [48]. The used liquid laundry detergents were: Maison DET (CHIMI-DET, Mahdia, Tunisia), Det, and New Det (Sodet, Sfax, Tunisia), Skip (Unilever, France), and Class (EJM, Sfax, Tunisia). The solid laundry detergents used were: Fino (FINO GmbH, Mangelsfeld, Germany), OMO (Unilever, France), Ariel (Procter & Gamble, Switzerland), EcoVax, Omino Bianco, Dipex, Dixan, and Nadhif (Henkel-Alki, Tunisia). Each enzyme was incubated with each detergent at 40°C for 1 h with 7 mg/mL of the above commercial detergents and the remining lipase activity was measured at an optimum temperature and pH for each enzyme used, using TC4 or TC8. The enzyme activity in the absence of any tested detergent and incubated under the assay similar conditions was considered as the control.

**Washing performance analysis of FAL.** The effectiveness of FAL as a detergent ingredient was tested on pieces of white cotton fabric (7 cm × 7 cm) that had been stained with tomato sauce, ketchup, or egg yolk. The stained fabric pieces were subjected to various wash cycles of treatments using a combination of faucet water, Class detergent (with a final concentration of 7 mg/mL), and FAL solution (at 500 U/mL). These treatments were carried out at 40°C by stirring the solution for 30 min in 100 mL Erlenmeyer flasks at 200 rpm. Next, the fabric pieces were double rinsed with distilled water and dried. They were then assessed visually to determine the efficacy of the lipase in stain removal. The extraction of triolein (TC18) or OO was carried out using a Soxhlet extractor and petroleum ether for 6 h. The quantity of OO that was eliminated was determined using the following equation [49]:

**Removal (%) = [Weight of total OO before washing (mg) − Weight of total OO after washing (mg)] / [Weight of total OO after washing (mg)] × 100**

**Statistical analysis.** The trials were conducted with a minimum of 3 separate experiments and the results were evaluated under conditions identical to the control, which lacked lipase. Data were expressed as the average of the results and their ± SD (standard deviation). Data analysis was performed with the Student–Newman–Keuls multiple comparison test and ANOVA. The results obtained were compared to identify any statistically significant differences. A *P*-value of less than 0.05 was used to determine the level of significance. The statistical analysis was performed with the 18.1.08 software.

**Culture collection depository's numbers and nucleotide sequence accession numbers.** The culture of the CBS isolate of the Actinomycota fungus was preserved in the Collection Tunisienne de Microorganismes CTM<TUN> in compliance with the Bacteriological Code (1990 revision) as amended by the International Code of Nomenclature of Prokaryotes (ICSP) at the plenary sessions in Sydney and Paris. The culture was kept at the Centre of

Biotechnology of Sfax (CBS) in Sfax, Tunisia, and at the Westerdijk Fungal Biodiversity Institute (WFDI) in Utrecht, the Netherlands, under the following authentic culture numbers: CTM 10621 and CBS: 21.09/Det, respectively. The sequences of the *ITS*, *TEF1-α*, and *RPB2* genes were registered in the GenBank/ENA/DDBJ databases with the following accession numbers: OQ122083, OQ181215, and OQ181216, respectively.

## Results

### Isolation and screening of lipase-producing fungi

The olive crops grow easily in Tunisia and in many other countries of the Mediterranean basin, thus occupying the largest agricultural area. Unfortunately, infection by *Fusarium* causes root rot in the olive tree. In this context, the diversity of fungi was discovered in two different olive groves located in two geographically distant governorates, Gabes (El Amarat) and Tataouine (Henchir El Ghazal) in the South-East of Tunisia, with different management practices. The fungi were isolated from the soil and from the major species of olive trees *Olea europaea* cv. Chemlali. A total of 67 fungal isolates were identified by phenotypic, biochemical, and phylogenetic analysis. 20 fungal species were identified belonging to eight different genera (*Aspergillus*, *Alternaria*, *Rhizoctonia*, *Verticillium*, *Macrophomina*, *Cladosporium*, *Penicillium*, and *Fusarium*) up to the point of monomorphic culturing, performed in solid medium and based on the screening of isolates on PDA plates containing OO and rhodamine B. Ten lipase-producing fungi (TN10, M45, AS15, AI16, R22, CBS, F6, P63, C58, and V35) were obtained showing lipase activity after 5 days of incubation at 25˚C (Fig 1). Only one, namely CBS, was selected and this showed a lipolytic activity on optimized culture medium of 25 U/mL on triglycerides (TGs), such as TC8, and 30 U/mL on an egg PC, using the pH-STAT technique (Fig 2).

### Fungus morphology

The colonies of fungal strain CBS grown on PDA had an average diameter of 55 mm ± 4 mm after 4 d-incubation in the dark at 25˚C. The development of the isolate was also evaluated on SNA after six days of incubation, and the average colony diameter was 83 mm ± 2 mm. The aerial mycelium observed on SNA was limited and had a pale white color, covering the entire surface of the medium in each culture plate. The morphology of the CBS isolate was compared to the initial description of *Fusarium annulatum* published by Bugnicourt (1952), as well as mentioned in Nelson et al. [38] and Yilmaz et al. [39] (Table 1).

The data revealed that the fungal strain CBS culture had a characteristic deep purple pigmentation on PDA (Fig 3). Typically, the aerial mycelium had a fluffy texture that initially appeared white but, as it matured, it acquired a purple hue and turned gray in certain spots. At times, dark purple sporodochia were observed. The rare macroconidia had thin and straight septations (3–4). The lengths of the macroconidia ranged from 15 μm to 56 μm and widths ranged from 1.9 μm to 3.5 μm. The apical cells of the macroconidia were rounded, and the basal cells were in the shape of a foot. Numerous microconidia were observed in long chains attached to monophylies and polyphylies. The microconidia had no septa, and their shapes wide-ranging from obovoid or closely oval with a truncated base to fusiform, with sizes ranging from 4.8 μm to 14.1 μm × 1.8 μm to 2.4 μm. No chlamydospores were present.

### Molecular and phylogenetic study of the isolated fungus

The PCR amplification of the *ITS*, *TEF1-α*, and *RPB2* genes produced fragments of 840 bp, 690 bp, and 937 bp, respectively. The CBS isolate showed identical results in all three replicates

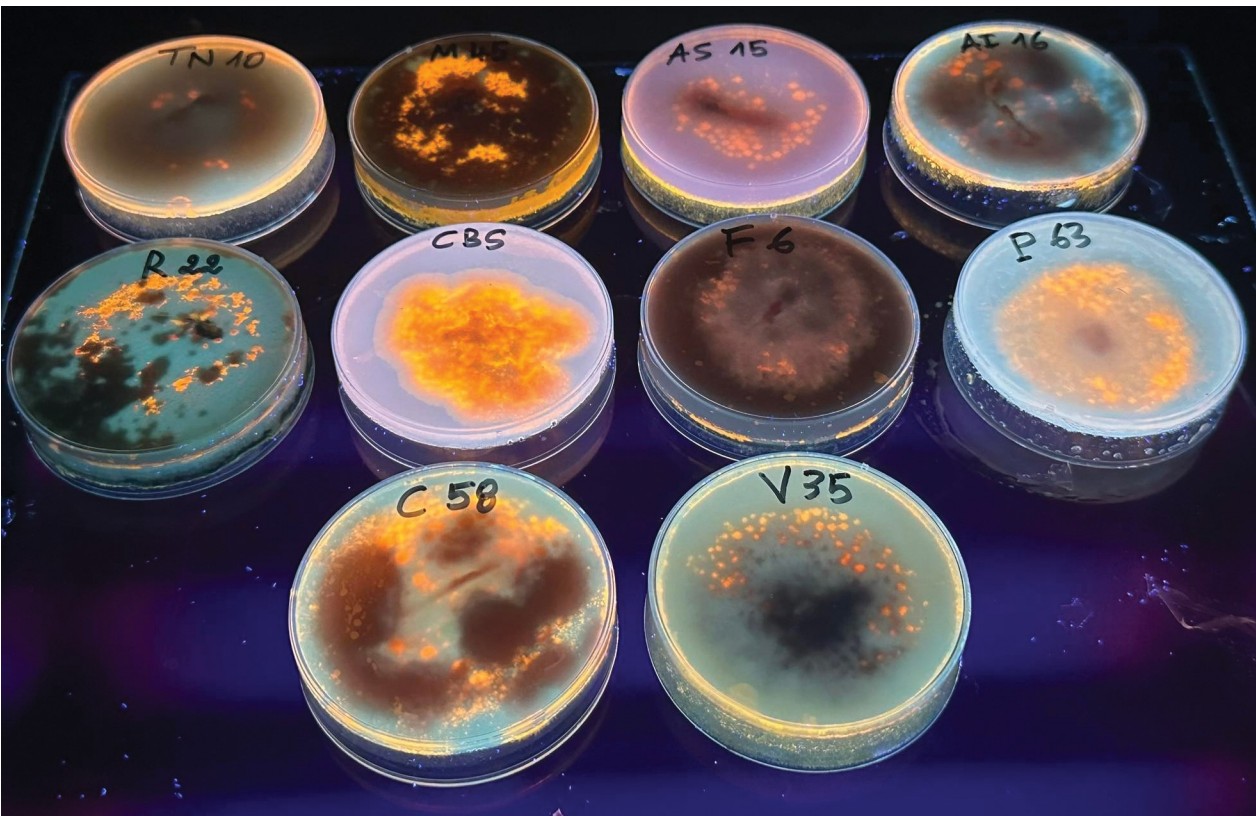

**Fig 1. Fluorescent haloes of lipase-producing fungi, on rhodamine B with 1% (*v/v*) olive oil agar medium, were visible under UV light at 365 nm.** Ten lipase-producing fungi (TN10, M45, AS15, AI16, R22, CBS, F6, P63, C58, and V35) were obtained showing variable lipase activities. After incubation for 5 days at 25˚C plates were subjected to UV irradiation and photographed.

and the nucleotide sequences were submitted to the GenBank database with accession numbers OQ122083, OQ181215, and OQ181216 for *ITS*, *TEF1-α*, and *RPB2*, respectively.

BLAST analysis was performed using GenBank (NCBI) which showed that the CBS isolate's *ITS* gene was completely (100%) matched to *Fusarium annulatum* (GenBank accession no.: MH862668). Additionally, the *TEF1-α* gene showed 99.28% sequence similarity with the MT010994 sequence and the *RPB2* gene had 99.47% sequence identity with the MT010983 sequence, both matching the *Fusarium annulatum* strain CBS 258.54. These results are presented in Table 2.

The CBS isolate was confirmed as belonging to the *Fusarium fujikuroi* species complex (FFSC) by a BLAST search of 3 sequences in the *Fusarium*-ID database [50]. To further verify its placement within the *Fusarium* genus, a combined analysis of the *ITS*, *TEF1-α*, and *RPB2* sequences was carried out in comparison with type strains belonging to the *Fusarium* section. The phylogenetic tree shows that the CBS isolate is part of a group of strains from the *Fusarium* section and it is maintained by 86% of the bootstrap samples (Fig 4).

The results of the phylogenetic analysis confirmed that the CBS isolate is *Fusarium annulatum*. The isolate was deposited in two culture collections, the "Collection Tunisienne de Microorganismes" (CTM) in Tunisia and the Westerdijk Fungal Biodiversity Institute (WFDI) in the Netherlands, with the numbers CTM 10621 and CBS: 21.09/Det, respectively. This was done in line with the guidelines and rules of the Bacteriological Code.

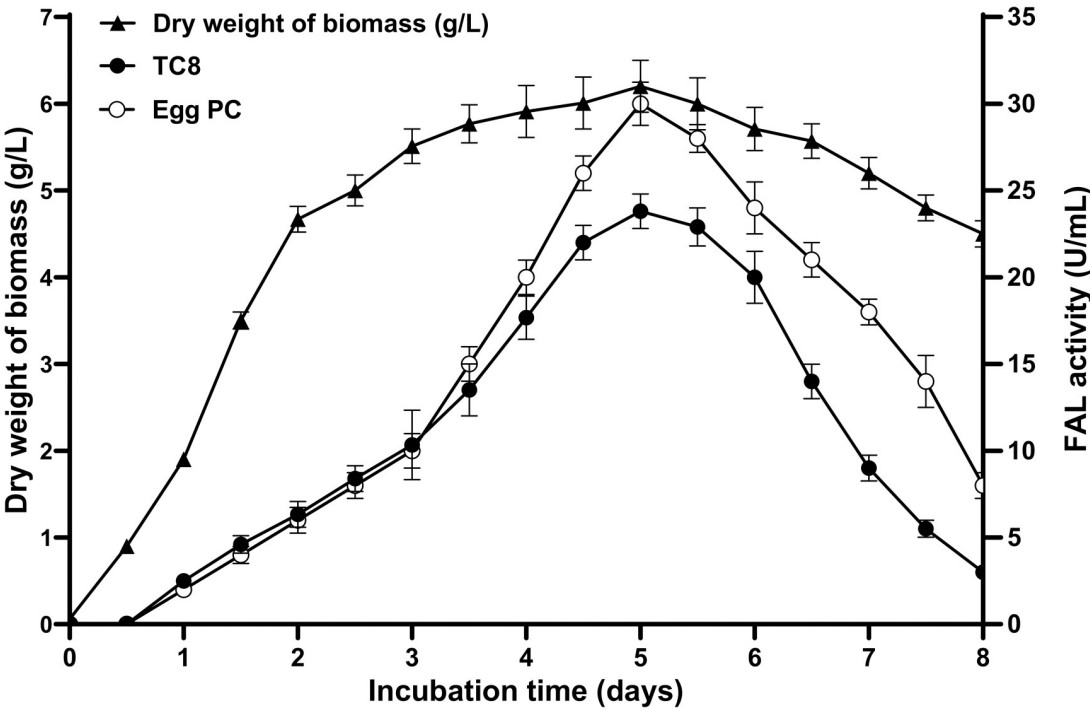

**Fig 2. Time course of the *Fusarium annulatum* Bugnicourt strain CBS.** Biomass (dry weight of mycelia) (π) and lipolytic enzyme activity using TC8 (●) or egg PC () as the substrate on optimized culture medium. Cultures were performed in 200 mL Erlenmeyer flasks of optimized liquid medium A containing: 15 g/L casein peptone, 5 g/L yeast extract, 1.75 g/L KH$_2$PO$_4$, 0.5 g/L MgSO$_4$, and 1% olive oil and incubated, at 25˚C, under agitation of 160 rpm. Cell growth was performed by measuring the dry matter. Lipase or phospholipase activity was determined in culture filtrates obtained after removal of cells by centrifugation, as described in the Methods section. Each point represents the mean (n = 3) ± standard deviation.

## Lipase production

Lipases are usually produced on carbon sources, such as oils, fatty acids, glycerol or Tweens, added with an organic nitrogen source [51,52]. Various initial tests were conducted to select the optimum components of the medium, such as the carbon source, phosphorus nitrates etc., and the optimum culture conditions (agitation and temperature of the medium). In the light of the results, a medium (medium A) consisting of (g/L): casein peptone (15), yeast extract (5), KH$_2$PO$_4$ (1.75), MgSO$_4$ (0.5), and 1% of OO (pH 6) with 160 rpm and 25˚C, was found to be the most suitable for production of lipase by the strain CBS. The production of lipase from strain CBS is induced by the presence of 1% OO. As illustrated in Fig 2, lipase production began on the initial day of cultivation and peaked at the end of the exponential growth phase, on the fifth day of cultivation. The maximum value attained was 25 U/mL or 30 U/mL, respectively, using TC8 or egg PC as substrates.

## Lipase purification

Throughout all the purification steps, TC8 was used as the substrate. The lipase from strain CBS was purified as reported in Section 4 (Section 4.7). The ammonium sulphate precipitate was loaded on to a Superdex® 200 Increase 10/300 GL gel filtration column and fractions containing lipase activity were further purified on HiTrap® Q-Sepharose FF columns by employing a linear gradient of NaCl (0–500 mM) in buffer A. The peak of lipase activity appeared between 180 mM and 200 mM NaCl (Fig 5A).

**Table 1. Morphological features of *Fusarium annulatum* isolate CBS, causing fruit rot in cantaloupes, compared to previous descriptions of *Fusarium annulatum* by Bugnicourt (1952), Nelson et al. [38] and Yilmaz et al. [39].**

| Morphological features | | *Fusarium annulatum* Bugnicourt (1952) grown on corn meal agar (CMA) | Isolate CBS grown on potato dextrose agar (PDA) |
|---|---|---|---|
| Colony features on PDA | Top view | Aerial mycelium absent or late-developed, very light, powdery, slightly dispersed, white in colour. Blackish purple pigments are normally formed in synthetic cultures | Aerial mycelium with cottony appearance, initially white orange, but gradually turning grayish violet with age. It even had a grey colouration in some areas |
| | Reverse view | ND | Accentuated purple colouration |
| Macroconidia | Shape | Thin-walled, strongly curved and sickle-shaped, with the basal cell clearly foot-shaped | Straight and slender; the apical cell was blunt and the basal cell foot-shaped |
| | Size: Length (μm) × Wide (μm) | 13–58 × 1.9–3.3 | 15–53 × 2.1–4.8 |
| | Number of septa | 3–6 | 3–5 |
| Microconidia | Shape | Cylindrical or claviform with a truncate tip | Ovoid or nearly ovoid with a truncate tip and, rarely, fusiform |
| | Size: Length (μm) × Wide (μm) | 4.7–14.4 × 1.7–2.3 | 4.8–13.5 × 1.6–2.7 |
| | Number of septa | 0–1 | 0–1 |
| Phialide | | Monophialide and simple polyphialides | Monophialide and simple polyphialides |
| Chlamydospores | | Absent | Absent |

ND, not described.

SDS-PAGE analysis of the active protein fraction showed a major protein band with a molecular mass of approximately 33 kDa, as well as a weak intensity protein band with a molecular mass of 40 kDa (Fig 5B, left, line 4; S1 File). Zymogram analysis of the purified

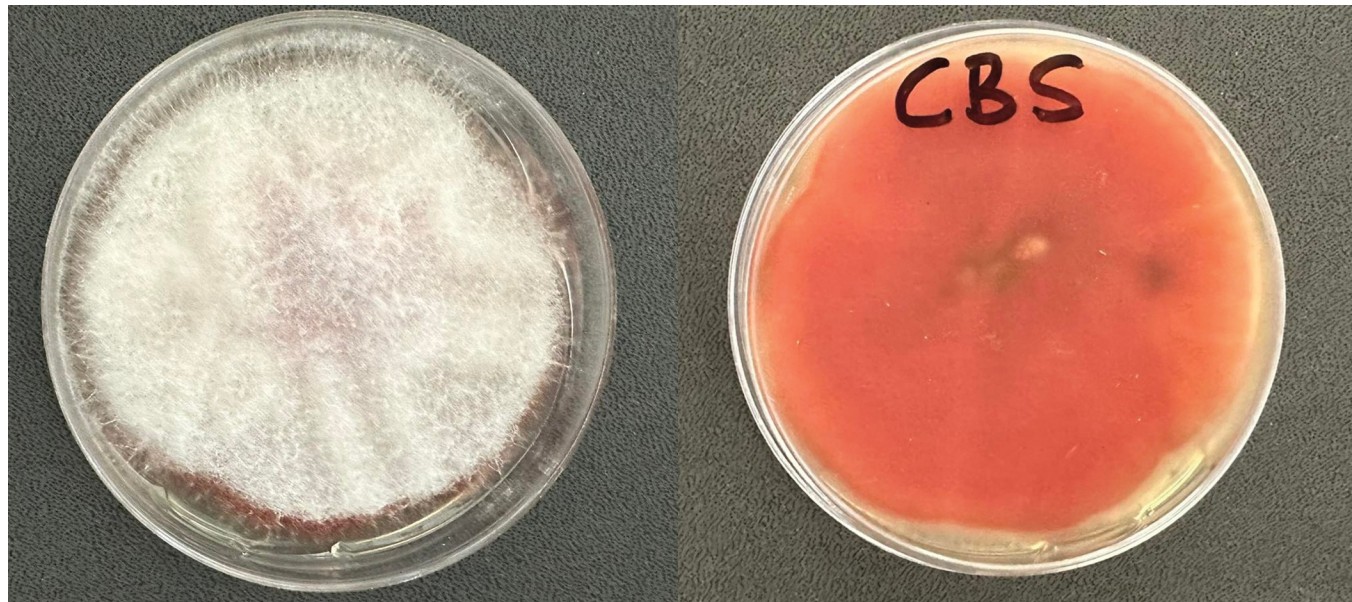

**Fig 3. Colony of strain CBS isolated from soil-borne fungi of the olive tree, *Olea europaea* cv. Chemlali.** (A) Upper view of a colony on PDA. (B) reverse view of colony on PDA.

**Table 2. Blast search for gene sequences of the *Fusarium annulatum* Bugnicourt strain CBS, comparted to the reference sequences obtained from type culture material.**

| GenBank accession no. (CBS) | DNA target | BLAST match sequence | | | |
|---|---|---|---|---|---|
| | | Reference Accession no. | Type material | Coverage (%) | Identity (%) |
| OQ122083 | *ITS*, rRNA[a] | *Fusarium annulatum* MH862668 | CBS 738.97 | 68 | 100.00 |
| | | *Fusarium annulatum* MH857317 | CBS 258.54 | 66 | 99.28 |
| | | *Fusarium concentricum* MH862659 | CBS 450.97 | 66 | 99.82 |
| | | *Fusarium fujikuroi* NR_111889 | CBS 221.76 | 66 | 99.82 |
| | | *Fusarium proliferatum* KR071678 | CBS 480.77 | 49 | 97.84 |
| | | *Fusarium globosum* LT746280 | CBS 431.97 | 87 | 100.00 |
| OQ181215 | *TEF1-α*[b] | *Fusarium annulatum* MT010994 | CBS 258.54 | 100 | 99.28 |
| | | *Fusarium concentricum* MT010992 | CBS 450.97 | 100 | 93.49 |
| | | *Fusarium fujikuroi* AB725605 | CBS 221.76 | 96 | 96.23 |
| | | *Fusarium proliferatum* KU604400 | CBS 480.77 | 95 | 97.82 |
| | | *Fusarium globosum* MW402131 | CBS 431.97 | 94 | 97.86 |
| OQ181216 | *RBP2*[c] | *Fusarium annulatum* MT010983 | CBS 258.54 | 99 | 99.47 |
| | | *Fusarium concentricum* MT010981 | CBS 450.97 | 99 | 97.86 |
| | | *Fusarium fujikuroi* KU604255 | CBS 221.76 | 84 | 96.85 |
| | | *Fusarium proliferatum* KU604245 | CBS 480.77 | 84 | 99.75 |
| | | *Fusarium globosum* MW402816 | CBS 431.97 | 82 | 99.48 |

[a] *ITS*, internal transcribed spacer; rRNA, ribosomal gene.

[b] *TEF1-α*, translation elongation factor 1-α gene.

[c] *RPB2*, second largest subunit of RNA polymerase gene.

fractions revealed a single active lipase band stained with fluorogenic lipase substrate 4-methylumbelliferone butyrate (MUFB) and this activity is associated with a 33 kDa protein (Fig 5B right, lines 2, 3, and 4; S2 File). The results of the FAL purification procedure are summarized in Table 3. A 62-fold purification was attained from the crude extract with a complete recovery of 21% (Table 3). A yield of approximately 0.16 mg of FAL, with a specific activity of 3500 U/mg, was obtained by using TC8 as the substrate from a 500 mL culture medium that was 5 days old.

## NH$_2$-terminal sequence determination of FAL

The first NH$_2$-terminal sequencing of FAL electroblotted onto a polyvinylidene fluoride (PVDF) membrane allowed the identification of 15 residues and these were shown to be E-T-C-M-D-K-G-S-K-V-T-E-W-T-V. BLAST searches revealed that this NH$_2$-terminal sequence displayed a high degree of identity (94%) with various uncharacterized proteins of the similar genus.

## Biochemical characterization of FAL

**Effects of pH on FAL activity and stability.** The enzyme activities of FAL were analyzed using TC8 or egg yolk PC as substrates at varying pH levels. The pH activity profile is illustrated in Fig 6A. FAL was shown to be highly active on TGs (TC8) between pH 8 and pH 10, with an optimal pH at 9. It is worth noting that at pH 7 and at pH 11, this TG lipase activity represents approximately 70% of the maximum activity at pH 9. This lipase activity was,

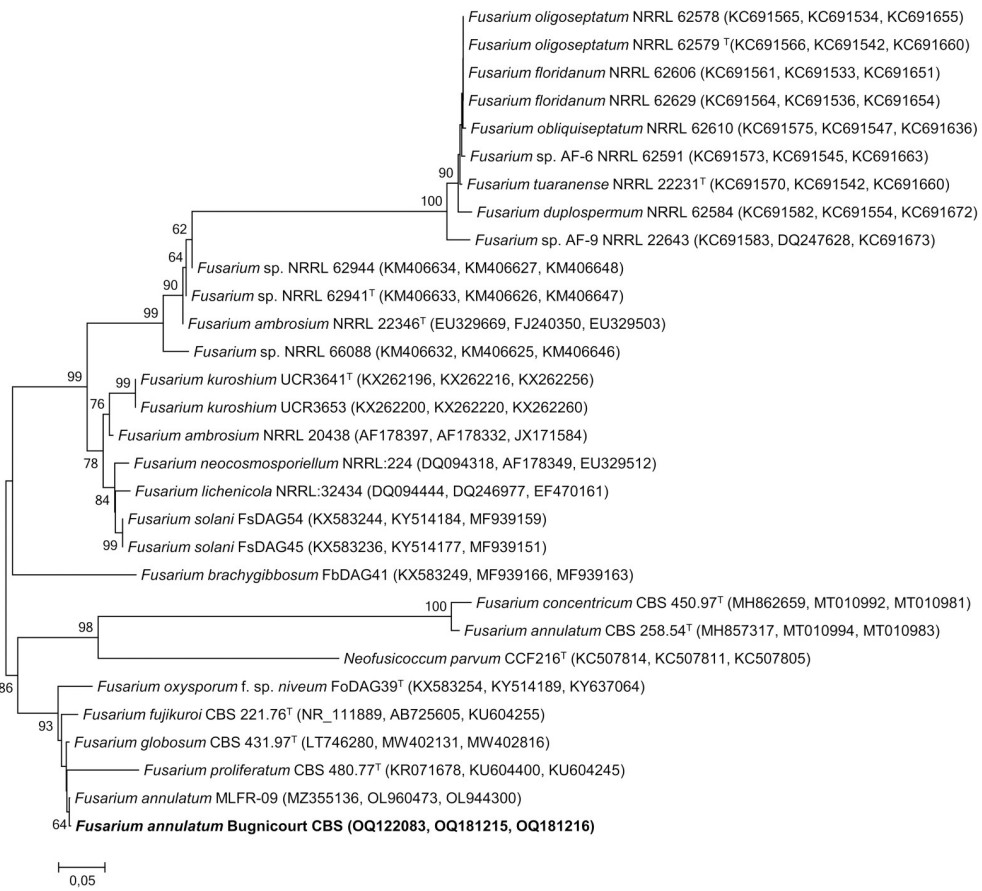

Fusarium oligoseptatum NRRL 62578 (KC691565, KC691534, KC691655)
Fusarium oligoseptatum NRRL 62579 T(KC691566, KC691542, KC691660)
Fusarium floridanum NRRL 62606 (KC691561, KC691533, KC691651)
Fusarium floridanum NRRL 62629 (KC691564, KC691536, KC691654)
Fusarium obliquiseptatum NRRL 62610 (KC691575, KC691547, KC691636)
Fusarium sp. AF-6 NRRL 62591 (KC691573, KC691545, KC691663)
Fusarium tuaranense NRRL 22231ᵀ (KC691570, KC691542, KC691660)
Fusarium duplospermum NRRL 62584 (KC691582, KC691554, KC691672)
Fusarium sp. AF-9 NRRL 22643 (KC691583, DQ247628, KC691673)
Fusarium sp. NRRL 62944 (KM406634, KM406627, KM406648)
Fusarium sp. NRRL 62941ᵀ (KM406633, KM406626, KM406647)
Fusarium ambrosium NRRL 22346ᵀ (EU329669, FJ240350, EU329503)
Fusarium sp. NRRL 66088 (KM406632, KM406625, KM406646)
Fusarium kuroshium UCR3641ᵀ (KX262196, KX262216, KX262256)
Fusarium kuroshium UCR3653 (KX262200, KX262220, KX262260)
Fusarium ambrosium NRRL 20438 (AF178397, AF178332, JX171584)
Fusarium neocosmosporiellum NRRL:224 (DQ094318, AF178349, EU329512)
Fusarium lichenicola NRRL:32434 (DQ094444, DQ246977, EF470161)
Fusarium solani FsDAG54 (KX583244, KY514184, MF939159)
Fusarium solani FsDAG45 (KX583236, KY514177, MF939151)
Fusarium brachygibbosum FbDAG41 (KX583249, MF939166, MF939163)
Fusarium concentricum CBS 450.97ᵀ (MH862659, MT010992, MT010981)
Fusarium annulatum CBS 258.54ᵀ (MH857317, MT010994, MT010983)
Neofusicoccum parvum CCF216ᵀ (KC507814, KC507811, KC507805)
Fusarium oxysporum f. sp. niveum FoDAG39ᵀ (KX583254, KY514189, KY637064)
Fusarium fujikuroi CBS 221.76ᵀ (NR_111889, AB725605, KU604255)
Fusarium globosum CBS 431.97ᵀ (LT746280, MW402131, MW402816)
Fusarium proliferatum CBS 480.77ᵀ (KR071678, KU604400, KU604245)
Fusarium annulatum MLFR-09 (MZ355136, OL960473, OL944300)
**Fusarium annulatum Bugnicourt CBS (OQ122083, OQ181215, OQ181216)**

**Fig 4. Molecular identification of the strain CBS.** Multilocus phylogenetic analysis using combined sequences from the *ITS*, *TEF1-α*, and *RPB2* gene regions showing the position of *Fusarium annulatum* Bugnicourt strain CBS (in bold) within the cluster comprising *Fusarium* species. Isolate CBS in bold was sequenced in this study. The sequences of *Neofusicoccum parvum* strain CCF216ᵀ (GenBank accession no.s: KC507814, KC507811, KC507805) were used as root, and the root position of the neighbor-joining tree was estimated using this strain as the outgroup. Distances and clustering were calculated using the neighbor-joining method. The tree topology of the neighbor-joining data was evaluated by Bootstrap analysis with 100 re-samplings. Bar, 0.05 substitutions per nucleotide position. Numbers at nodes (>50%) indicate support for the internal branches within the tree obtained by bootstrap analysis (percentages of 100 bootstraps). GeneBank accession numbers are presented in parentheses. T: Type strain, same formatting.

however, significantly reduced at pH 5 (30% of the maximum activity, Fig 6A). Interestingly, FAL exhibited maximum PLA₁ activity in a rather pH narrow range of 9–11 with an optimal pH at 11 (Fig 6A) and, therefore, it can be considered to be an alkaline enzyme. To determine the pH stability of FAL, the enzyme was pre-incubated at room temperature (25±2°C) over a broad pH range of 4–12 for 1 h. As shown in Fig 6B, FAL exhibited exceptional stability at pH values from 5 to 11 and preserved 100% of its maximal activities between pH 8 and 9, on TC8, and 75%-100% of its maximal activities between pH 4 and 12 on egg PC.

**Effect of temperature on FAL activity and stability.** FAL activity was evaluated under the standard assay conditions at temperatures ranging from 25°C to 55°C. The data show that FAL exhibited maximum TG lipase and PLA₁ activity at 40°C and 45°C, respectively (Fig 7A).

The thermostability profile of FAL showed that the lipase was able to retain its full activity at relatively high temperatures (above 40°C). The TG lipase (Fig 7B) and PLA₁ (Fig 7C) activities of FAL were maintained at 100% after incubation of the enzyme at 30°C, 37°C, or 40°C for 1 h. While the TG lipase activity of FAL is maintained at 100% afterward 1 h-incubation of the

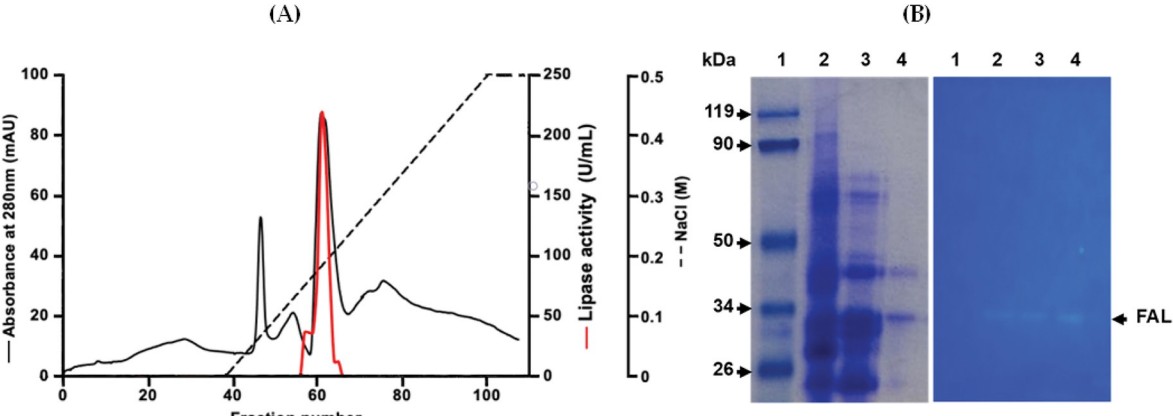

**Fig 5. Purification and electrophoretic analysis of FAL.** (**A**) Chromatogram profile of FAL purification on a HiTrap™ Q-Sepharose FF column. Adsorbed proteins were eluted with a linear NaCl gradient of 0 to 0.5 M NaCl in buffer A. FAL activity was measured, as described in Material and Methods (Section 4.5), using TC8 as the substrate. (**B**) SDS-PAGE (12%) analysis of eluted proteins. Lane 1, molecular mass marker; lane 2, resuspended pellets after ammonium sulphate (80%) precipitation; line 3, fraction obtained after gel filtration chromatography on Superdex® 200 Increase 10/300 GL column; lines 4 and 5, purified fractions (7 μg) from HiTrap™ Q-Sepharose FF column. (**C**) FAL activity staining with MUF-butyrate.

enzyme at 45˚C (Fig 7B), $PLA_1$ activity is reduced by 20% of maximum $PLA_1$ activity under the same conditions (Fig 7C).

**Effect of metal ions on FAL activity.** Chelating reagents, such as EDTA and EGTA, were found to inhibit FAL activity, indicating that metal cofactors are critical for enzyme function (Table 4). Consequently, the impact of different divalent ions on FAL activity was examined. As seen in Table 4, incubation resulted in an increase in lipase activity (1 mM) with $Ca^{2+}$, $Mn^{2+}$, or $Mg^{2+}$ (176%, 150%, or 116%, respectively), compared to the control. However, $Co^{2+}$ and $Fe^{2+}$ decreased the FAL activity by 92% and 34%, respectively (Table 4). Some other heavy metal ions ($Ni^{2+}$, $Hg^{2+}$, and $Cd^{2+}$) completely inactivated FAL while $Zn^{2+}$ inhibited the enzyme activity by 60% (Table 4). A slight inhibitory effect (10%) on enzyme activity was observed with $Ba^{2+}$. No effect on FAL activity was observed with $Cu^{2+}$.

The influence of various $Ca^{2+}$ concentrations on FAL activity has been investigated (Fig 8A). In the absence of $Ca^{2+}$ and in the presence of 10 mM of EDTA (Fig 8A), only 57% of the TG lipase activity (Table 4) and 6% of the $PLA_1$ activity were observed. With the addition of $Ca^{2+}$, TG lipase activity increased, reaching a maximum of 3500 U/mg using TC8 as the substrate at 40˚C and pH 9 in the presence of 2–3 mM $CaCl_2$ (Fig 8A). The TG lipase activity decreases slightly thereafter to reach a plateau at 3500 U/mg in the presence of 4–8 mM $CaCl_2$

**Table 3. Flow sheet for the purification of the lipase FAL from *Fusarium annulatum* Bugnicourt strain CBS.**

| Purification Steps[a] | Total Activity (U)[b] | Total Protein (mg)[c] | Specific Activity (U/mg) | Purification (-fold) | Yield (%) |
|---|---|---|---|---|---|
| Crude extract | 11900 ± 34 | 225 ± 4 | 53 ± 2 | 1 | 100 |
| $(NH_4)_2SO_4$ precipitation (80%) | 5800 ± 15 | 8 ± 1 | 725 ± 5 | 14 | 49 |
| Superdex® 200 Increase 10/300 GL column | 3750 ± 10 | 1.5 ± 0 | 2500 ± 7 | 53 | 32 |
| HiTrap™ Q-Sepharose FF column | 560 ± 3 | 0.16 ± 0 | 3500 ± 9 | 62 | 21 |

[a] Experiments were conducted, from 500 mL of culture, at least three independent times and ± standard errors (SE) are reported.

[b] One unit; micromole of fatty acid released per min using TC8 as the substrate of lipase activity.

[c] Amounts of protein were determined as described elsewhere.

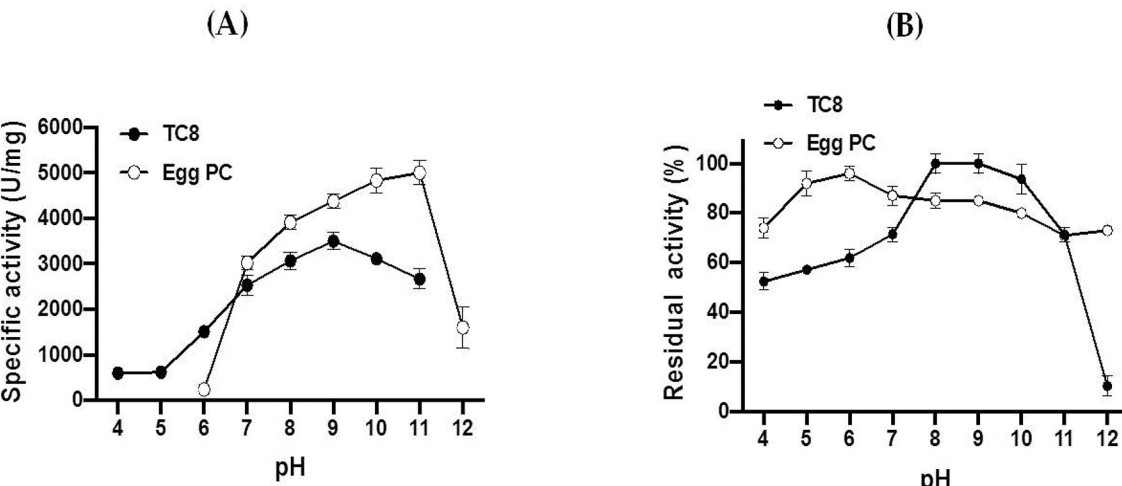

**Fig 6.** Effects of pH on the activity (A) and stability (B) of the purified FAL using TC8 or egg PC at 40˚C under standard conditions. The pH profile was determined in different buffers by varying the pH values from 4 to 12. The pH stability of the FAL was determined by incubating the enzyme at different pH values, ranging from 4 to 12, for 1 h at 40˚C and the residual activity was measured at pH 9 on TC8 or pH 11 on egg PC, also at 40˚C. The maximum activity on TC8, obtained at pH 9, or on egg PC, at pH 11, was considered as 100%. Each point represents the mean of three independent experiments.

(Fig 8A). Interestingly, the PLA$_1$ activity of FAL increased with increasing Ca$^{2+}$ concentrations, reaching a plateau at 5000 U/mg in the presence of 4 mM CaCl$_2$ (Fig 8A).

**Effect of bile salts, inhibitors and chemical reagents on FAL activity.** Knowing that surface active agents are frequently used in lipase assays, potentially increasing lipase activity through interfacial activation [53], we examined the effect of the NaTDC on the FAL activity. As exposed in Fig 6B, the presence of bile salts seems to enhance the TG lipase activity of FAL up to a limiting concentration of 1 mM, to reach its maximum specific activity of 3500 U/mg. Nevertheless, outside 1 mM of NaTDC, the TG lipase activity decreases rapidly, down to 60% and 20% of the residual activity at 2 mM and 5 mM NaTDC, respectively (Fig 8B). The inhibition exerted on the lipase activity of FAL by NaTDC, at concentrations greater than 1 mM, might be due to the fact that NaTDC prevents the binding of the enzyme at the lipid substrate interface. Regarding the PLA$_1$ activity of FAL, the enzyme has a low PLA$_1$ activity (410 U/mg) in the absence of NaTDC and this activity increases gradually with the increase of NaTDC concentrations, reaching a maximum activity (5000 U/mg) at 3 mM NaTDC. Beyond 3 mM of NaTDC, the PLA$_1$ activity of the enzyme decreases slightly until stabilizing at 90% of the maximal activity (Fig 8B).

To examine whether FAL is a serine enzyme, the impact of orlistat on FAL activity during the lipolysis of TC8 (as illustrated in Fig 9A) or egg PC (as depicted in Fig 9B) was studied. Orlistat, a potent digestive lipase inhibitor, binds covalently to the catalytic site serine. Injection of Orlistat (40 μM, final concentration) at 8 min (Fig 9A, red curve) and 8 min (Fig 9B, red curve) completely inhibited the TG lipase and PLA$_1$ activity of FAL, respectively. As a control, DMSO injection alone had no effect on FAL activity (Fig 9, blue curve). In addition, other serine-modifying reagents, such as PMSF and DIFP, completely inhibited FAL activity (Table 4).

As shown in Table 4, FAL was marginally affected by thiol reagents. FAL retained 70%, 90%, 102%, or 97% of its original activity when treated for 60 min with DTNB (10 mM), NEM (2 mM), iodoacetamide (5 mM), or PAO (10 mM), respectively (Table 4). Treatment of FAL

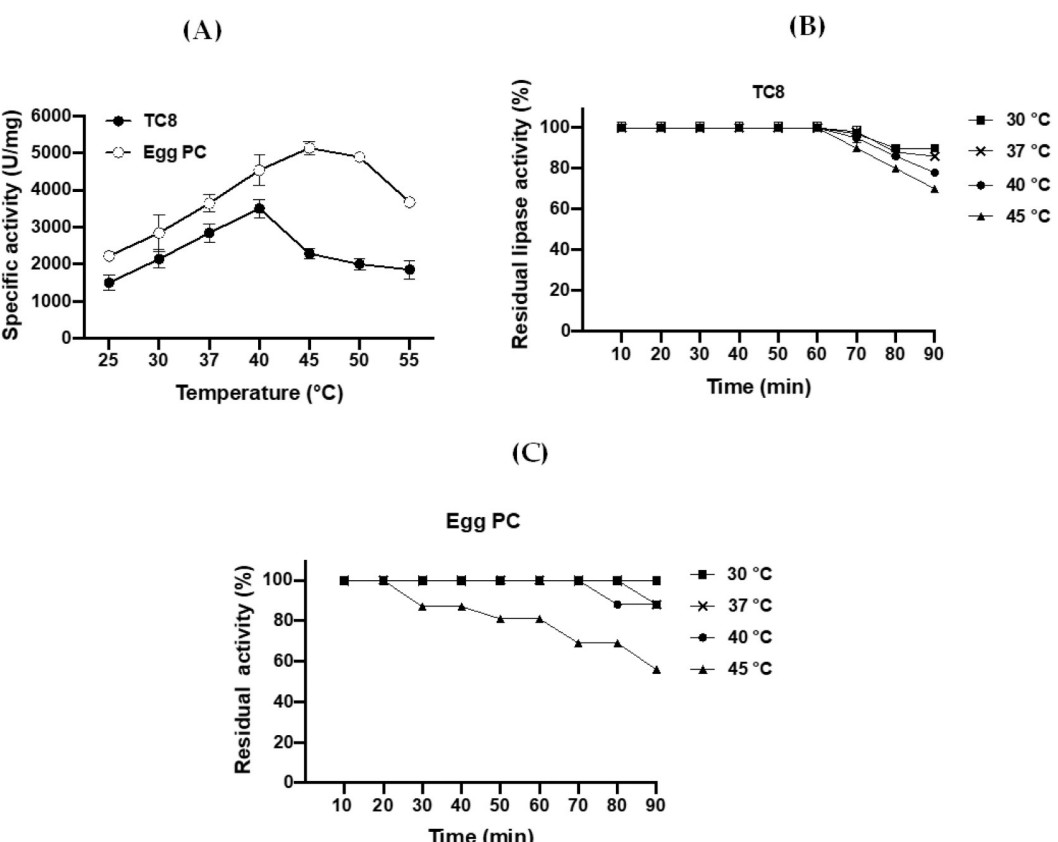

**Fig 7. Effects of temperature on the activity and stability of the purified FAL using TC8 or egg PC at 40°C under standard conditions.** (**A**) Enzymatic activity at various temperatures (from 25°C to 55°C) was determined using TC8 or egg PC at pH 9 under standard conditions. For stability of FAL on TC8 (**B**) at pH 9 or egg PC (**C**) at pH 11, the activity was measured after incubation of the enzyme for the indicated time at various temperatures under standard conditions. Each point represents the mean of three independent experiments.

with reducing agents (10 mM), such as 2-ME or DL-DTT), for 60 min had no significant effect on catalytic activity (Table 4) with residual activities of 98% or 96%, respectively (Table 4).

**Substrate specificity of FAL.**   The substrate specificity of FAL was determined using TGs of varying acyl chain length and egg PC as substrates (Fig 10A). FAL hydrolyzes short to medium chain TGs with optimal activity on TC8 (3500 U/mg) followed by TC2 (2800 U/mg), TC4 (2400 U/mg) and TC6 (2000 U/mg) (Fig 10A). The activity of FAL on OO emulsion represents approximately 60% (1800 U/mg) of that on TC8 (Fig 10A). Interestingly, FAL is able to efficiently catalyze the hydrolysis of phospholipids, in the presence of a concentration of 4 mM $CaCl_2$, 3 mM of NaTDC and at pH 11 and 45°C, with a specific activity of 5000 U/mg on egg PC.

To examine the regioselectivity of FAL for the PC *sn*-1 or *sn*-2 positions, the hydrolysis degree of the surface-coated *sn*-EOPC esterified at the *sn*-1 position or *sn*-OEPC esterified at the *sn*-2 position was determined (Fig 10B). It should be noted that each substrate contains only one ester bond and one non-hydrolyzable ether bond with a non-UV-absorbing alkyl chain, in order to prevent acyl chain migration during lipolysis. Upon injecting FAL onto coated *sn*-EOPC, there was a rapid increase in absorbance at 272 nm, indicating the enzyme's high $PLA_1$ activity. In contrast, injecting FAL onto coated *sn*-OEPC did not result in any hydrolysis, thus confirming the enzyme's high $PLA_1$ activity (Fig 10B).

**Table 4. Effects of some selected metal ions, inhibitors and chemical reagents on the purified FAL from *Fusarium annulatum* Bugnicourt strain CBS.** The enzyme assay was performed after pre-incubation of the enzyme with each tested chemical compound, for 1 h at 40°C. The non-treated and dialyzed enzyme was considered as 100% for the metal ion assay. The lipase activity measured in the absence of any inhibitor or reducing agent was taken as the control and considered as 100%. Residual activity was measured at pH 9 and 40°C, using TC8 as a substrate.

| Metal Ions/Inhibitors/Chemical Reagents [a] | Concentration (mM) [a] | Residual Lipase Activity (%) [b] |
|---|---|---|
| None | – | 100 [b] ± 3 |
| $Ca^{2+}$ ($CaCl_2$) | 1 | 176 [a] ± 6 |
| $Fe^{2+}$ ($FeSO_4$) | 1 | 66 [bc] ± 2 |
| $Mn^{2+}$ ($MnCl_2$) | 1 | 150 [a] ± 5 |
| $Mg^{2+}$ ($MgCl_2$) | 1 | 116 [b] ± 4 |
| $Ba^{2+}$ ($BaCl_2$) | 1 | 90 [b] ± 3 |
| $Zn^{2+}$ ($ZnSO_4$) | 1 | 40 [cd] ± 1 |
| $Cu^{2+}$ ($CuCl_2$) | 1 | 100 [b] ± 3 |
| $Co^{2+}$ ($CoCl_2$) | 1 | 8 [d] ± 0 |
| $Ni^{2+}$ ($NiCl_2$) | 1 | 0 |
| $Hg^{2+}$ ($HgCl_2$) | 1 | 0 |
| $Cd^{2+}$ ($CdCl_2$) | 1 | 0 |
| None | – | 100 [b] ± 3 |
| PMSF | 5 | 0 |
| DIFP | 2 | 0 |
| Benzamidine | 2 | 103 [b] ± 3 |
| DTNB | 10 | 69 [bc] ± 2 |
| NEM | 2 | 90 [b] ± 2 |
| Iodoacetamide | 5 | 102 [b] ± 3 |
| PAO | 10 | 97 [b] ± 2 |
| 2-ME | 10 | 98 [b] ± 2 |
| DL-DTT | 10 | 96 [b] ± 2 |
| EDTA | 10 | 57 [b] ± 2 |

[a–d] Means in the same column of each parameter with different lower-case letters differed significantly ($P < 0.05$). Incubation with the purified enzyme at 40°C for 1 h. Values represent the means of three independent replicates, and ± SE are shown.

## Enzymatic performance of the purified FAL

**Effect of organic solvents on FAL stability.** In this study, the stability of FAL in the presence of various water miscible and immiscible organic solvents (Log $P > 1.8$) was assayed and compared to that of *Gibberella zeae* Lipase (GZEL), an extracellular enzyme secreted by *Fusarium oxysporum* [34], and Lipolase®, a commercially available lipase. The results were analyzed in terms of residual activity relative to an untreated control. FAL was found to be very stable when incubated, for 24 h, at a final concentration of 25% (*v/v*) of the following non-polar organic solvents (Log $P > 1.8$)—cyclohexane, *n*-hexane, *n*-hexadecane, toluene, *n*-decane, chloroform, *n*-hexanol and isooctane, retaining 189%, 165%, 135%, 111%, 105%, 102%, 95%, and 90% of its initial activity, respectively, (Fig 11). Under the same experimental conditions, GZEL and Lipolase® show, relatively speaking, the same stability as FAL in these solvents, with the exception of *n*-hexadecane and *n*-decane which reduce Lipolase® activity by 34% and 23%, respectively (Fig 11). However, FAL is shown to be less stable when incubated with protic polar solvents. In the presence of isopropanol, methanol, n-butanol or ethanol, FAL retained 80%, 56%, 55%, or 42% of its activity, respectively (Fig 11) whilst GZEL retained 91%, 92%,

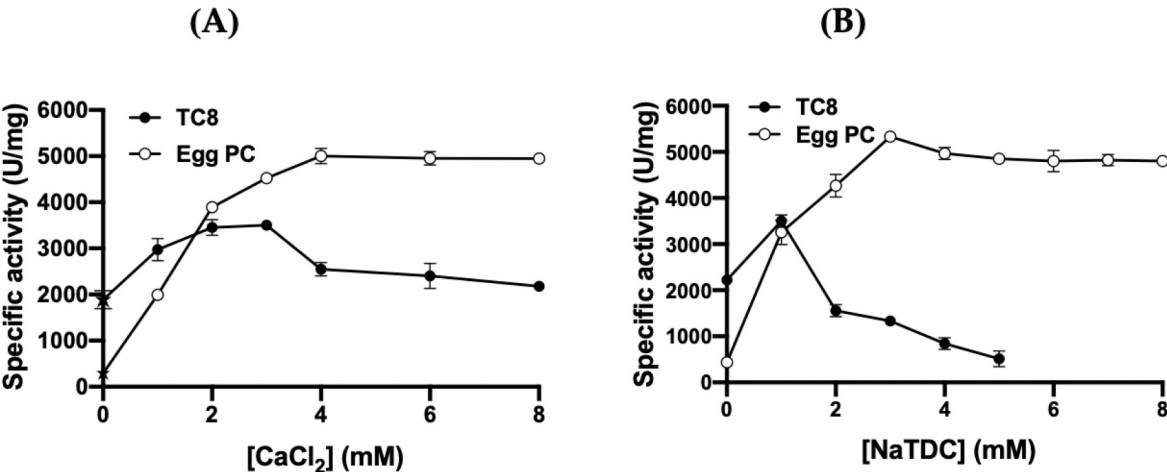

**Fig 8. Influence of Ca²⁺ (CaCl₂) and NaTDC on FAL activity.** (**A**) Effect of the concentration of Ca²⁺ on FAL activity. Enzyme activity was measured at increasing concentrations of Ca²⁺. TC8 emulsions as substrate for lipase activity in the presence of 2 mM NaTDC, and PC as substrate for PLA₁ activity in the presence of 4 mM NaTDC. (**B**) Effect of increasing concentrations of bile salt (NaTDC) on lipase activity in the presence of 2 mM CaCl₂ and phospholipase activity in the presence of 4 mM CaCl₂, using TC8 emulsion and phosphatidylcholine as substrates, respectively. The star indicates the FAL activity measured in the absence of CaCl₂ and in the presence of 10 mM EDTA. Each point represents the mean of three independent experiments.

62%, or 51% of its activity, respectively (Fig 11). Lipolase®, under the same experimental conditions retained, respectively, 91%, 155%, 58%, or 120% of its initial activity (Fig 11).

The effect of aprotic polar solvents was also tested on enzyme activity. FAL is shown to be very stable when incubated with dimethylformamide (DMF) or DMSO with residual activities of 105% or 110%, respectively. DMF or DMSO also have a positive effect on the stability of GZEL and Lipolase®, with remining activities, respectively, of 93% or 101% for GZEL and 120% or 88% for Lipolase®. It should be noted that the further aprotic polar organic solvents, acetonitrile and ethyl acetate, both had a negative effect on the activity of FAL, GZEL and Lipolase®, with remining activities of 66% or 35% for the FAL, 77% or 16% for GZEL and 55% or 15% for Lipolase® (Fig 11).

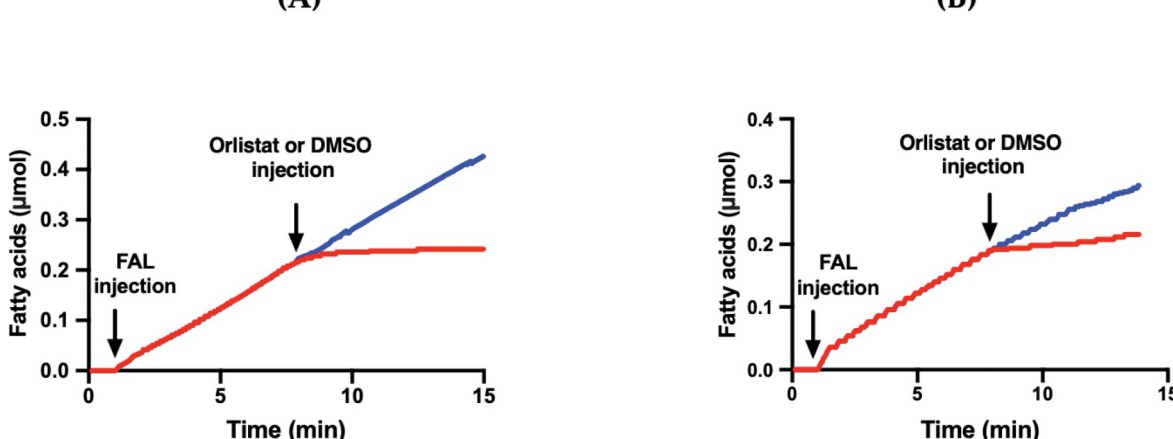

**Fig 9. Effect of Orlistat on the FAL activities.** DMSO (blue lines) or Orlistat in DMSO (40 μM, final concentration, red curves) was injected into the reaction medium at 8 min (A) and 8 min (B) after starting lipolysis with FAL on TC8 and on egg PC, respectively. Lipase and PLA₁ activities were measured, at pH 9 and 40°C and at pH 11 and 45°C, using TC8 and egg PC as substrates, respectively. Curves are representative of three independent experiments.

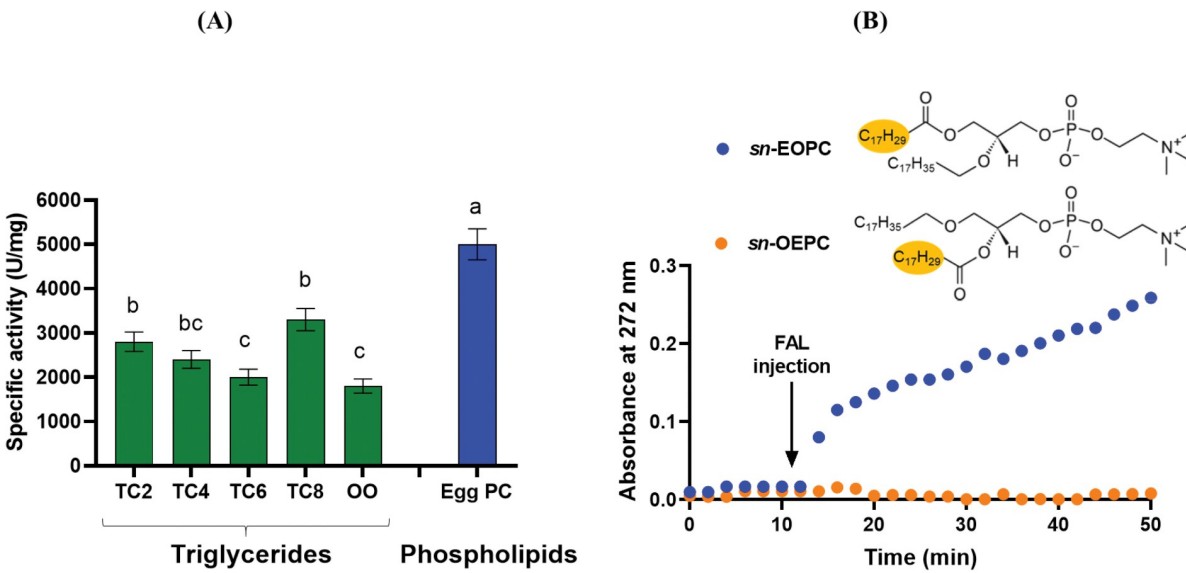

**Fig 10. Substrate specificity of FAL.** (**A**) Chain length selectivity of FAL. The specific activity was measured under standards conditions using TC2, TC4, TC8, olive oil (OO), or egg PC as substrate, as described in Material and Methods. (**B**) Kinetic recordings of coated *sn*-EOPC or *sn*-OEPC lipolysis by FAL. Variations, with time, of the absorbance at 272 nm were recorded for 10 min, for stabilization, and then for 40 min after FAL injection (0.65 µg per well).

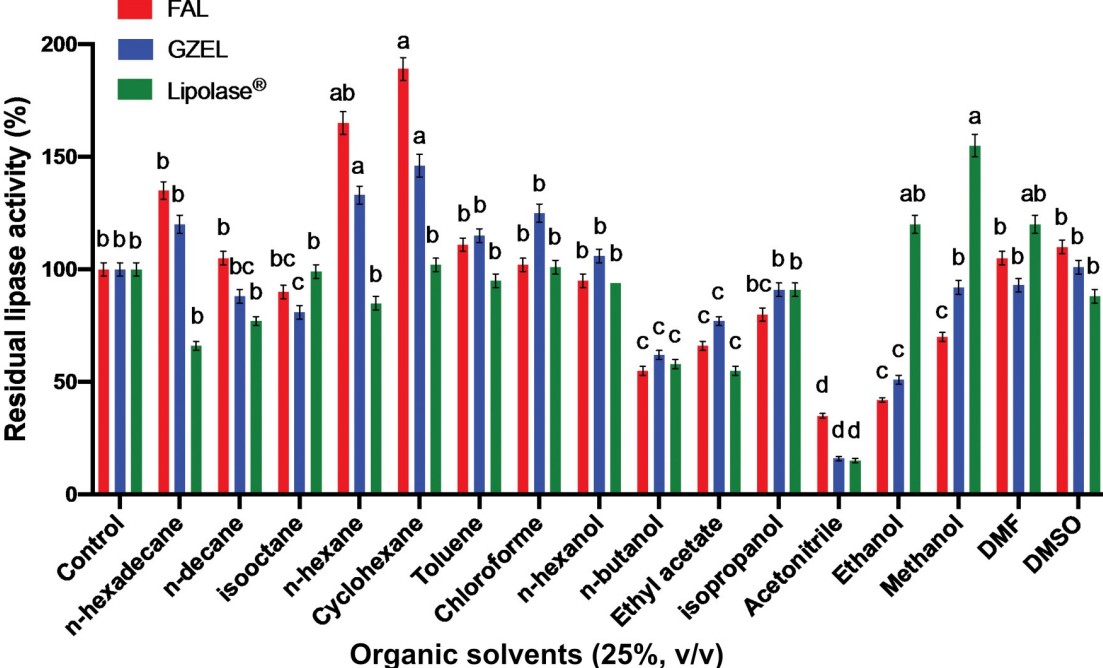

**Fig 11. Effect of organic solvents on the activity and stability of FAL, GZEL, and Lipolase®.** The effect of organic solvents was determined by incubating each enzyme with the solvent [25% (*v/v*) final concentration] for 24 h. The residual lipase activities were determined under the same conditions, using TC8 as the substrate at 40°C and pH 9, as described in the Materials and Methods, and then expressed as a percentage of the activity level in the absence of organic solvents. The activity of the enzyme without any organic solvent was taken as 100%. Each point represents the mean of three independent experiments. Vertical bars indicate standard error of the mean (*n* = 3). [a–d] Means in indicator enzymes with different lowercase letters differed significantly ($P < 0.05$).

**Table 5. Effects of various laboratory detergent additives on the stability of FAL compared to GZEL and Lipolase®.**

| Detergent Additives | Final Concentration | Residual Lipase Activity (%) [a,b] | | |
|---|---|---|---|---|
| | | FAL | GZEL | Lipolase® |
| Control | 0 | 100 [cd] ± 3 | 100 [cd] ± 3 | 100 [cd] ± 3 |
| Zeolite | 1% (w/v) | 99 [cd] ± 3 | 95 [cd] ± 2 | 86 [cd] ± 2 |
| STPP | 0.2% (w/v) | 155 [ab] ± 4 | 138 [ab] ± 4 | 150 [ab] ± 4 |
| | 0.5% (w/v) | 95 [cd] ± 2 | 90 [cd] ± 2 | 93 [cd] ± 2 |
| SDS | 1% (w/v) | 135 [a] ± 4 | 97 [cd] ± 2 | 96 [cd] ± 2 |
| | 2% (w/v) | 110 [bc] ± 3 | 86 [cd] ± 2 | 75 [d] ± 2 |
| $Na_2CO_3$ | 50 mM | 182 [a] ± 4 | 151 [a] ± 4 | 154 [ab] ± 4 |
| | 100 mM | 152 [ab] ± 4 | 144 [ab] ± 4 | 126 [bc] ± 3 |
| Tween 20 | 1% (v/v) | 93 [cd] ± 2 | 88 [cd] ± 2 | 73 [d] ± 2 |
| | 5% (v/v) | 52 [de] ± 1 | 77 [d] ± 2 | 54 [de] ± 1 |
| Tween 40 | 1% (v/v) | 107 [bc] ± 3 | 96 [cd] ± 2 | 87 [cd] ± 2 |
| | 5% (v/v) | 36 [e] ± 1 | 39 [e] ± 1 | 36 [e] ± 1 |
| Tween 60 | 1% (v/v) | 80 [cd] ± 2 | 90 [cd] ± 2 | 98 [cd] ± 2 |
| | 5% (v/v) | 28 [e] ± 1 | 61 [de] ± 2 | 60 [de] ± 2 |
| Tween 80 | 1% (v/v) | 80 [cd] ± 2 | 91 [cd] ± 2 | 90 [cd] ± 2 |
| | 5% (v/v) | 33 [e] ± 1 | 48 [de] ± 1 | 50 [de] ± 1 |
| $Na_2CMC$ | 1% (w/v) | 128 [ab] ± 4 | 95 [cd] ± 2 | 92 [cd] ± 2 |
| | 10% (w/v) | 104 [c] ± 3 | 65 [de] ± 2 | 52 [de] ± 1 |
| Triton X-100 | 1% (v/v) | 147 [ab] ± 4 | 125 [cd] ± 2 | 85 [cd] ± 2 |
| | 5% (v/v) | 111 [bc] ± 3 | 107 [bc] ± 3 | 68 [de] ± 2 |
| TAED | 0.5% (w/v) | 145 [ab] ± 4 | 131 [bc] ± 3 | 116 [bc] ± 3 |
| | 5% (w/v) | 105 [cd] ± 3 | 96 [cd] ± 2 | 75 [d] ± 2 |
| Sodium perborate | 1% (v/v) | 122 [ab] ± 3 | 102 [ab] ± 3 | 113 [bc] ± 3 |
| | 5% (v/v) | 105 [bc] ± 3 | 74 [b] ± 2 | 88 [cd] ± 2 |
| $H_2O_2$ | 5% (v/v) | 150 [ab*] ± 4 | 133 [ab*] ± 3 | 80 [cd*] ± 2 |
| | 10% (v/v) | 124 [ab*] ± 4 | 106 [bc*] ± 3 | 66 [de*] ± 2 |

[a–e] Means in the same column of each parameter with different lower-case letters differed significantly ($P < 0.05$). The lipase was incubated for 1 h with a detergent additive, at 40°C, and the residual lipase activity was determined under optimal assay conditions for each enzyme. Data presented are the average of at least 3 sets of tests, ± SE. * Data collected in the presence of 100 mM borate-NaOH buffer.

**Effects of detergent additives on the stability of FAL and GZEL.** The stability of a lipolytic enzyme in the presence of detergent ingredients is a crucial characteristic in choosing a good detergent lipase. For this reason, the stability of FAL and GZEL were examined and compared to that of Lipolase® by incubating the enzyme with detergent additives, which were either commercially available (Table 6) or commonly found in laboratories (Table 5), for 1 h. As Table 5 reveals, the FAL was more stable and exhibited 145%, 128%, 182%, or 155% of its initial activity with the following anti-redeposition agents TAED (0.5%, w/v), Na₂CMC (1%, w/v), Na2CO3 (100 mM), or STPP (0.2%, w/v), respectively (Table 5). In the same experimental conditions, GZEL exhibited 131%, 95%, 151%, or 138% of its initial activity, respectively (Table 5). For oxidants, FAL activity, with 5% or 10% (v/v) H₂O₂ or 1% or 5% (v/v) sodium perborate, was increased by 150% and 124% or 122% and 105%, respectively, while GZEL exhibited 133% and 106% or 102% and 74% of its initial activity, respectively (Table 5). FAL showed considerable stability towards strong anionic surfactant as it retained 110% (Table 5), 93% or 125% (Table 6) of its initial activity with SDS (2%, w/v), Galaxy LAS (1%, v/v), or

**Table 6. Effects of various commercialized detergent additives on the stability of FAL compared to GZEL and Lipolase®.**

| Commercialized Detergent Additives | Concentration (%, *v/v*) | Residual Lipase Stability (%) [a,b] | | |
|---|---|---|---|---|
| | | FAL | GZEL | Lipolase® |
| Control | 0 | 100 [cd] ± 3 | 100 [cd] ± 3 | 100 [cd] ± 3 |
| SAFOL 23 E7 | 0.5 | 156 [ab] ± 4 | 140 [ab] ± 4 | 142 [ab] ± 4 |
| | 1 | 109 [bc] ± 3 | 102 [cd] ± 3 | 90 [cd] ± 2 |
| Dehydol® LT 7 | 0.5 | 162 [ab] ± 4 | 138 [ab] ± 4 | 164 [a] ± 4 |
| | 1 | 98 [cd] ± 2 | 94 [cd] ± 2 | 108 [bc] ± 3 |
| SURFAC® LM 30 | 0.5 | 191 [a] ± 5 | 146 [ab] ± 4 | 120 [bc] ± 3 |
| | 1 | 160 [ab] ± 4 | 108 [bc] ± 3 | 94 [cd] ± 2 |
| NEODOL® 25–7 | 0.5 | 95 [cd] ± 2 | 97 [cd] ± 2 | 87 [cd] ± 2 |
| | 1 | 44 [ef] ± 1 | 51 [de] ± 1 | 45 [e] ± 1 |
| Galaxy LAS | 2 | 93 [cd] ± 2 | 92 [cd] ± 2 | 95 [cd] ± 2 |
| | 5 | 50 [de] ± 1 | 56 [de] ± 2 | 59 [de] ± 1 |
| Galaxy LES 70 | 2 | 125 [cd] ± 3 | 113 [bc] ± 3 | 94 [cd] ± 2 |
| | 5 | 75 [d] ± 2 | 66 [de] ± 2 | 50 [de] ± 1 |
| Galaxy 110 | 2 | 129 [cb] ± 3 | 120 [bc] ± 3 | 125 [cb] ± 3 |
| | 5 | 96 [cd] ± 2 | 94 [cd] ± 2 | 83 [cd] ± 2 |
| Galaxy CAPB Plus | 1 | 112 [bc] ± 3 | 133 [b] ± 3 | 14 [f] ± 4 |
| | 5 | 97 [cd] ± 2 | 99 [b] ± 3 | 105 [c] ± 4 |
| TERGITOL™ NP-9 SURFACTANT | 2 | 96 [cd] ± 2 | 78 [cd] ± 2 | 89 [cd] ± 2 |
| | 5 | 50 [ed] ± 1 | 47 [e] ± 1 | 50 [de] ± 1 |
| FINDET® AR/52 | 0.5 | 85 [cd] ± 2 | 80 [cd] ± 2 | 75 [d] ± 2 |
| | 1 | 46 [ef] ± 1 | 45 [ef] ± 1 | 28 [ef] ± 1 |
| Anti-foam | 0.5 | 94 [cd] ± 2 | 97 [cd] ± 2 | 88 [cd] ± 2 |
| | 1 | 76 [cd] ± 2 | 68 [de] ± 2 | 50 [ed] ± 1 |
| Formol | 0.2 | 180 [ab] ± 5 | 165 [ab] ± 4 | 135 [a] ± 3 |
| | 0.5 | 154 [ab] ± 4 | 111 [bc] ± 3 | 84 [cd] ± 2 |
| Tinopal® CBS-X | 0.5 | 163 [ab] ± 4 | 151 [ab] ± 4 | 122 [bc] ± 3 |
| | 1 | 107 [bc] ± 3 | 105 [bc] ± 3 | 86 [cd] ± 2 |
| Sulfacid K | 10 | 70 [d] ± 2 | 80 [cd] ± 2 | 83 [cd] ± 2 |
| | 15 | 25 [f] ± 1 | 30 [ef] ± 1 | 25 [ef] ± 1 |
| Marlipal® 31/90 | 0.5 | 90 [cd] ± 2 | 70 [de] ± 2 | 83 [cd] ± 2 |
| | 1 | 48 [ef] ± 1 | 38 [ef] ± 1 | 35 [ef] ± 1 |
| EDTA | 0.2 | 141 [b] ± 3 | 122 [bc] ± 3 | 125 [bc] ± 3 |
| | 0.5 | 106 [c] ± 3 | 103 [c] ± 3 | 62 [de] ± 2 |
| Perfume I Class | 0.5 | 140 [ab] ± 4 | 153 [ab] ± 4 | 146 [ab] ± 4 |
| | 1 | 105 [c] ± 3 | 102 [b] ± 3 | 99 [cd] ± 2 |
| Perfume II Class | 0.5 | 160 [a] ± 4 | 144 [ab] ± 4 | 155 [ab] ± 4 |
| | 1 | 120 [bc] ± 3 | 115 [bc] ± 3 | 96 [cd] ± 2 |
| Propyl betaine | 1 | 106 [c] ± 3 | 92 [cd] ± 2 | 81 [cd] ± 2 |
| NaOH 50% | 1 | 121 [bc] ± 3 | 107 [c] ± 3 | 119 [bc] ± 3 |

[a–f] Means in the same column of each parameter with different lower-case letters differed significantly ($P < 0.05$). The lipase was incubated for 1 h with detergent additives, at 40°C, and the residual activity was determined under optimal assay conditions for each enzyme. Data presented are the averages of at least 3 sets of tests, ± SE.

Galaxy LES 70 (2%, *v/v*), respectively. Whereas GZEL and Lipolase® retained only 86% and 75% (Table 5) of their initial activity, respectively, in the presence of 2% (*w/v*) SDS.

Moreover, after treatment with non-ionic surfactants, such as Tweens (20, 40, 60, and 80) at 1% (*v/v*), FAL was shown to be relatively stable by conserving 93%, 107%, 80%, or 80% of its initial activity, respectively (Table 6). Similar results were observed with GZEL and Lipolase®. However, treatment of the enzyme with higher concentrations (5%, *v/v*) of these surfactants decreased the residual activity to 30%-50% for FAL, 40%-77% for GZEL and 36%-50% for Lipolase® (Table 5). FAL and GZEL were shown to be highly stable when treated with Triton-X100 (5%, *v/v*) with remining activities of 111% and 107%, respectively, while Lipolase® retained only 85% or 68% of its initial activity in the presence of 1% or 5% (*v/v*) of Triton X-100 (Table 5).

Additionally, residual activity of 156%, 162%, or 191% was observed in the presence of Dehydol®LT 7, SAFOL 23E7 or SURFAC® LM 30, as commercial detergent additives, at 0.5% (*v/v*), respectively (Table 6). Similar results were observed for GZEL, with residual activity of 140%, 138%, or 146%, respectively and for Lipolase®, with residual activity of 142%, 164%, and 120%, respectively (Table 6).

The FAL exhibited remarkable stability, retaining 121%-180% residual activity in the presence of several other commercially available detergent additives, such as 0.5% Tinopal® CBS-X, 0.5% Formol, 0.5% Perfume I Class, and 1% Perfume II Class 0.2% EDTA (Table 6). GZEL also exhibited good stability in the presence (*v/v*) of 5% $H_2O_2$, 0.5% Formol or 0.5% Tinopal® CBS-X and retained 133% (Table 6), 111% or 151% activity (Table 6), respectively, *vs* Lipolase® which retained 80%, 84%, or 122% (Table 6), respectively.

**Stability and compatibility of lipases with various commercial laundry detergents.** In order to investigate the potential use of FAL as a bio-detergent additive, its stability and compatibility with commercially available detergent formulations were measured and compared to those of GZEL and Lipolase®. As shown in Fig 12, FAL was found to be highly stable and compatible with all tested commercial liquid and solid laundry detergents in which it retained 75%-100% of its original activity vs 72%-100% for GZEL and Lipolase®. FAL retained 100% of its initial activity with Class, Maison Det and New det while GZEL retained 97%, 96%, and 88% of its initial activity, respectively, and Lipolase® retained 92%, 88%, and 77% of its initial activity, respectively. GZEL was completely stable in the presence of Fino, Dixan, and Omino Bianco while Lipolase® was fully stable with Ariel, OMO, and X-Clean, retaining 100% of its initial activity (Fig 12). These findings supplied further support for the usefulness of FAL as a cleaning additive in bio-detergent formulations in future industrial applications.

**Wash performance test on oil removal with lipases used.** The effect of various commercial detergents on oil removal, in the absence or presence of a lipase solution, is illustrated in Fig 13. Irrespective of the detergent used, the addition of the lipase increases the cleaning efficiency of oil removal, which demonstrates the benefit of including this enzyme in the detergent composition. The effectiveness of the lipase depends on the detergent used. For example, FAL is more efficient in Nadhif, EcoVax, New Det, Fino, and Skip, increasing efficiency by 32%, 41%, 23%, 36%, and 50%, respectively, compared to 16%, 25%, 11%, 21%, and 35%, respectively for GZEL and 8%, 26%, 10%, 18%, and 37% respectively for Lipolase® (Fig 13). With other detergents, such as Class, Omino, Maison Det, Pro-Clean, and Axion, the addition of FAL or GZEL provides a greater improvement (44%-56%) than the addition of Lipolase® (11%-38%) (Fig 13). In some other detergents, like Dixan, GZEL provides a 67% improvement *vs* 39% for FAL and 38% for Lipolase®. In OMO it is Lipolase® which is more effective, providing a 60% improvement *vs* 48% and 50% for FAL and GZEL, respectively (Fig 13).

The importance of using lipases as a bio-additive in commercial detergents was established by visual examination of the removal of oil stains from cotton fabric. As shown in Fig 14, the

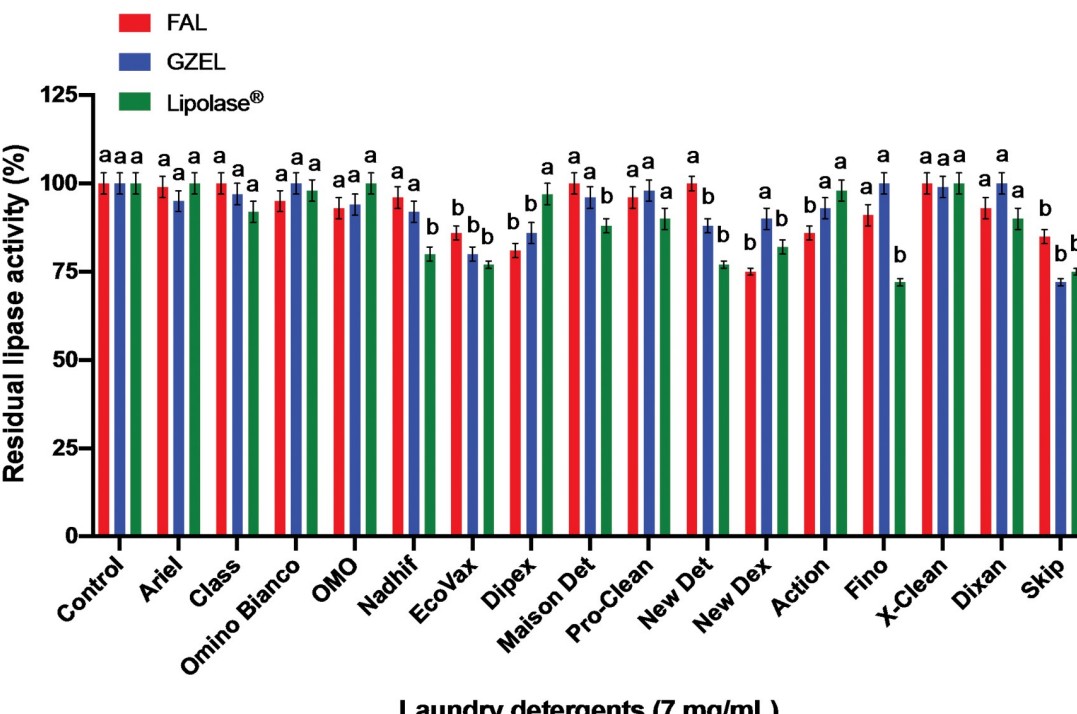

**Fig 12. Stability and compatibility of the tested lipases (FAL, GZEL, and Lipolase®) in the presence of commercial laundry detergents.** Lipases were incubated with laundry detergents (7 mg/mL) for 60 min at 40°C. The enzyme activity of the control sample, without additive and incubated under similar conditions, was taken as 100%. Each point represents the mean of three independent experiments. Vertical bars indicate the standard error of the mean ($n = 3$). [a–b] Means in indicator enzymes with different lowercase letters differed significantly ($P < 0.05$).

use of Class detergent supplemented with FAL, GZEL, or Lipolase® significantly increases the removal of the oily component from tomato sauce, ketchup, and egg yolk stains. In particular, FAL and GZEL, through their dual TG lipase and PLA$_1$ activity, are much more effective in removing oil stains from cotton fabric than the Lipolase® currently used.

## Discussion

Presently, the most severe diseases posing a threat to olive trees (*Olea europaea* cv. Chemlali) in Tunisia and, presumably, in many Mediterranean basin countries [30,32,33], are dieback and wilting symptoms brought on by a group of soil-borne fungi [26–29]. *Fusarium* is among the key phytopathogenic genera linked to dieback symptoms in olive trees. Following pathogenic testing on young olive trees (cv. Chemlali), it was determined that, out of 104 isolates of *Fusarium* spp., 23 were pathogenic while the remainder demonstrated weak or no pathogenicity. As far as we know, this is the first research to suggest that *Fusarium annulatum* could be a significant contributor to the dieback disease of olive trees in Tunisia. Additionally, this study sought to isolate a new fungal strain from a relatively extreme Tunisian biotope in the hope of finding an interesting new lipase producing strain. Based on morphological and molecular analyses, the selected strain CBS was identified as *Fusarium annulatum* Bugnicourt. For the molecular identification, the DNA region encompassing the *ITS1*, *5.8S*, and *ITS2* of ribosomal DNA was sequenced. This region has been identified as a suitable target for analysing fungal phylogeny, making it an appropriate choice for this study [54,55].

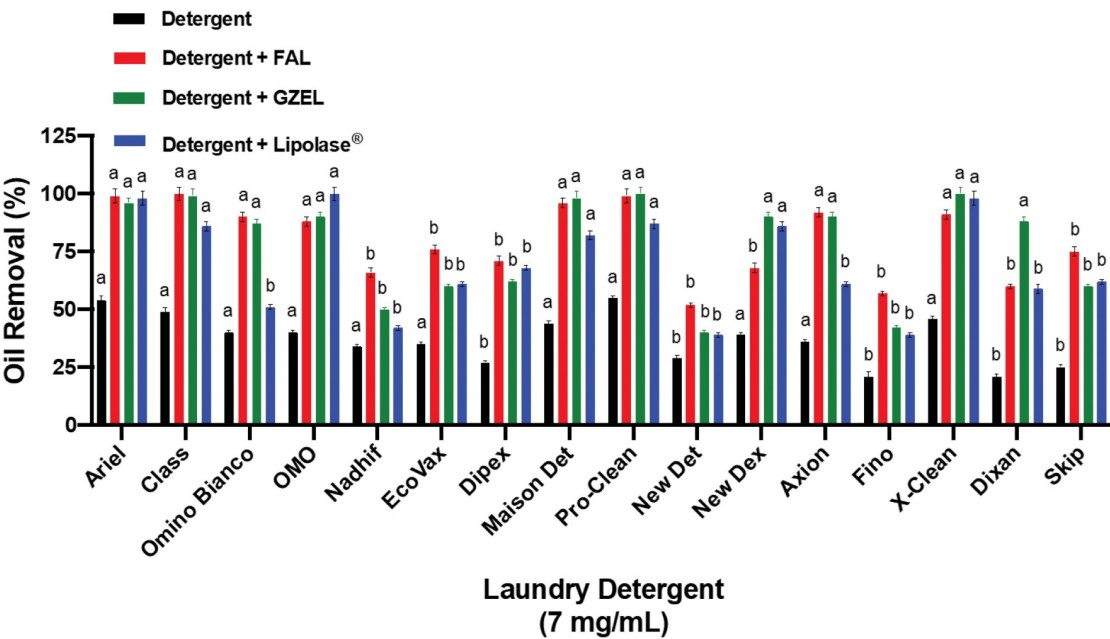

**Fig 13. Evaluation assay of used lipases on olive oil removal from cotton fabric with detergents.** Effect of FAL, GZEL, and Lipolase[®] on the removal of olive oil from cotton fabric with various liquid and solid laundry detergents. [a–b] Means for all columns of each parameter with different lower-case letters differed significantly ($P < 0.05$). Values represent the means of 3 independent replicates and the ± SE is shown.

Lipases are usually produced using oils, fatty acids, glycerol or tweens as carbon sources [51,52]. Similar to the reports for several *Fusarium* lipases [56–58], the novel extracellular lipase of this *Fusarium annulatum*, named FAL, has a molecular mass of 33 kDa as confirmed by SDS-PAGE and zymography analysis. This molecular mass is similar to that reported for the lipases of *Fusarium solani* [57] and *Fusarium oxysporum* [58]. However, the NH$_2$-terminal sequence of FAL presents high identity with numerous *Fusarium* uncharacterized proteins which have only been partially sequenced (around 18 kDa). Consequently, it was not possible to find the characteristic nucleophilic elbow highly conserved motif (GXSXG, where X can be any amino acid) which includes the catalytic serine residue found in the lipase family. It would be interesting to perform molecular cloning of the *lipFA* gene encoding the FAL protein in order to determine the full protein sequence.

Our results reveal that an optimal temperature for FAL is 40˚C—45˚C which is consistent with the *Fusarium verticillioides* [59] and *Fusarium graminearum* [60] lipases. Various fungal lipases have been reported to exhibit an ideal temperature from 37˚C to 50˚C, such as *Aspergillus* sp. ST11 lipase with an optimal activity at 37˚C [61], *Galactomyces geotrichum* lipase with an optimal activity at 45˚C [62], and *Talaromyces thermophilus* lipase showing maximum activity at 50˚C [63]. Regarding the influence of temperature on FAL stability, FAL was able to keep most of its catalytic activity at relatively high temperatures (above 40˚C). These findings are within the range of the results for a lipase previously studied, *Fusarium oxysprum* [64]. Furthermore, the FAL is more stable than the *Fusarium solani* lipase which, like the majority of fungal lipases, exhibits a higher temperature sensitivity at 45˚C [57].

The striking feature of FAL is its high alkaline pH stability and detergent stability. It is worth mentioning that most fungal and yeast lipases exhibit their maximum activity at a neutral or alkaline pH. Similarly, this lipase attains its peak activity within a pH range of 9 to 11, indicating that it is an alkaline enzyme. Under similar experimental conditions, Jalouli et al.

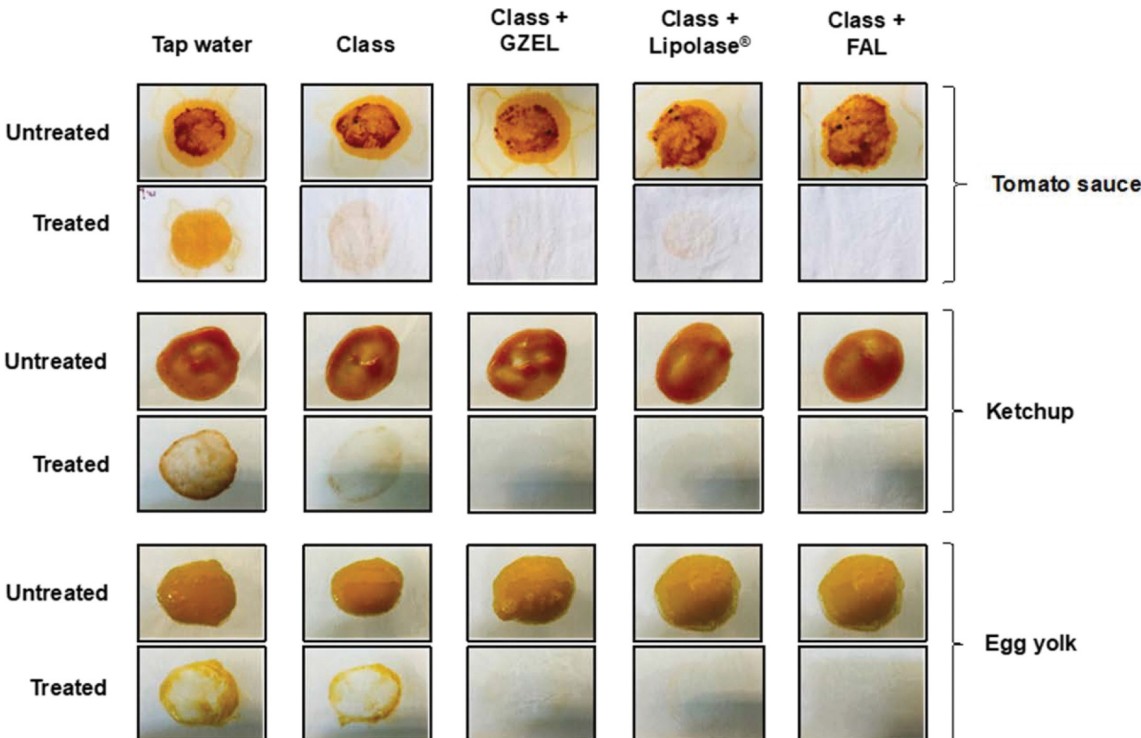

**Fig 14. Wash performance test on oil removal with lipases used.** Wash performance test on oil removal with a commercial detergent in the presence of FAL, GZEL, or Lipolase®. The washing performance analysis test of the lipase was conducted with the commercial detergent Class (7 mg/mL) using cotton cloth stained with tomato sauce, ketchup, or egg yolk. The oil-stained cloth was rinsed with tap water, washed with Class (7 mg/mL), washed with Class supplemented with GZEL (500 U/mL), washed with Ariel supplemented with Lipolase® (500 U/mL), or washed with Ariel supplemented with FAL (500 U/mL).

[57] signalled that the extreme lipase and PLA activity of lipase from *Fusarium solani* was observed at pH 8.5, a level of activity that decreases drastically at pH 6 [57]. Instead, Wang et al. [34] described that the extracellular lipase secreted by *Fusarium graminearum* (kwown as *Gibberella zeae*), named GZEL, exhibited maximum TG lipase and $PLA_1$ activity over a variety of pH 5–6 and this started to decrease at pH 9.

Our results confirmed that FAL has an exceptional pH stability spectrum at pH values from 5 to 11. Interestingly, this differs from those described in the literature for the majority of fungal lipases which are only stable within a narrow pH array from 4 and 7 and demonstrate less tolerance to alkaline pH levels [65], as seen for the lipases of *Rasamsonia emersonii* [66], *Aspergillus niger* [67], and *Penicillium aurantiogriseum* [68]. *Fusarium solani* lipase has been reported to be stable between pH 8 and 9 [57] and the elevated activity and stability at alkaline pH makes FAL an appropriate candidate for use in the manufacturing of detergents. Such alkaline enzymes are already used in heavy-duty detergents and dishwashing laundry [69–71].

The FAL activity has demonstrated notable stability with the chelating agents, making it an attractive choice for use as an additive in detergents. This is particularly advantageous as substantial quantities of chelating agents are added to improve stain removal and act as water softeners [72,73]. A FAL activity increase was detected with $Ca^{2+}$, $Mg^{2+}$, and $Mn^{2+}$, while $Fe^{2+}$ and $Co^{2+}$ reduced it. Other studies on *Fusarium solani* lipases have also reported similar results [57,74] where $Mn^{2+}$ and $Mg^{2+}$ have been found to be the primary enhancers of activity, while $Fe^{2+}$ and $Co^{2+}$ have been shown to depress activity. Mehta et al. [75] demonstrated that $Co^{2+}$ inhibited the extracellular lipase activity from *Aspergillus fumigatus*. The inhibition of several

lipases by metal ions may be due to the interaction between these ions and the thiol groups of cysteine residues located close to the active site of the enzyme [76]. However, a considerable reduction in the $PLA_1$ activity of FAL was seen without $Ca^{2+}$. Similar findings were reported for a (phospho)lipase from *Peziza* sp. [77] which requires the presence of $Ca^{2+}$. It seems that $Ca^{2+}$ could have three different functions in the action of lipases: removal of fatty acids as insoluble $Ca^{2+}$ salts, direct enzyme activation derived from the enzyme concentration at the lipid–water interface, and a structure stabilizing effect on the enzyme [45]. The behaviour of FAL with $Ca^{2+}$ resembles that of the known calcium dependent group of phospholipases $A_2$ ($PLA_2$) [78]. The structural analyses of $PLA_2$ have revealed the existence of a conserved loop for binding $Ca^{2+}$ in the 3D structures of these enzymes. Nevertheless, one study has indicated that $Ca^{2+}$ ions are not indispensable for a lipase that demonstrates $PLA_1$ activity [79]. To obtain a better insight of the impact of calcium ions on the $PLA_1$ activity of this FAL, additional structural analyses are needed.

The inhibition exerted on the TG lipase activity of FAL by NaTDC (>1 mM), may be due to the fact that the NaTDC maintains the binding of the FAL enzyme to the lipid-substrate interface. Comparable results have been obtained with the *Fusarium solani* lipases [80,81] as well as some other bacterial lipases, e.g., the lipases of *Serratia* sp. strain W3 [82], *Staphylococcus simulans* [83], and *Staphylococcus aureus* [84]. The $PLA_1$ activity of FAL is shown to be low (410 U/mg) without NaTDC and gradually increases to reach a maximum activity (5000 U/mg) at 4 mM in NaTDC. This stimulating impact on the $PLA_1$ activity, already noted for numerous phospholipases [77,85], may be explained firstly by its impact on the quality of the water-lipid interface tension by solubilization of the substrate phospholipids and/or the reaction product. Indeed, the emulsification effect of bile salts can promote substrate availability at a quantity around its critical micellar concentration [86,87]. Secondly, increasing bile salt concentration can lead to conformational changes rendering the protein more active [88]. Moreover, even at a high concentration of NaDTC (5 mM) the $PLA_1$ activity remains stable, as has been reported for phospholipase C from *Bacillus thuringensis* [85] and a lipase from *Fusarium solani* [57].

Orlistat is known to be a potent inhibitor of digestive lipases, and carboxylester hydrolases in general, that covalently binds to the seryl residue at the catalytic site [89]. It was observed that Orlistat totally inhibits TG lipase and PLA1 activities, indicating that a covalent complex is formed between the β-lactone ring of orlistat and the hydroxyl of the catalytic seryl residue of FAL. This mechanism has been previously established for digestive lipases, suggesting that Orlistat may also be an effective inhibitor of FAL [89]. In addition, FAL was found to be completely inactivated in the presence of other serine-reactive reagents (PMSF and DFP), providing further evidence that FAL is a serine hydrolase having a catalytic triad consisting of Ser-Asp(Glu)-His at the active site. TG lipase and $PLA_1$ activities were inhibited by serine-reactive reagents is an indication that it is the same catalytic site that is probably involved in the hydrolysis of TGs and phospholipids. Similar results were found with the *Fusarium solani* [57] and *Staphylococcus hyicus* [90] (phospho)lipases, which have the similar catalytic site for TG lipase and PLA activities.

One of the most interesting properties of FAL is that it can efficiently catalyze the hydrolysis of TGs of different acyl chain lengths and phospholipids. It should be noted that the $PLA_1$ activity is much higher (5000 U/mg) than that obtained previously with *Fusarium solani* lipase (2400 U/mg at pH 8.5), through the pH-STAT procedure and egg PC as the substrate [57]. The present findings demonstrate that FAL is a $PLA_1$ with a clear selectivity for the *sn*-1 position of phospholipids, using tailor-made substrates that distinguish between $PLA_1$ and $PLA_2$ activities [91]. We showed that FAL favourably hydrolyzes medium chain TGs (TC8), as declared for *Fusarium solani* [57,74], *Penicillium camembertii* Thom PG-3 [38] and *Penicillium cyclopium*

[42] lipases. According to Simons et al. *Staphylococcal* (phospho)lipases exhibit a broad range of substrates, encompassing TGs of different chain lengths, phospholipids, and lysophospholipids. The specific activities of these enzymes on TC4 and egg PC are reported to be 28 U/mg and 172 U/mg, respectively [92]. Furthermore, Ishibashi et al [93] recently identified a new lipase/phospholipase from a Traustochytrid, known as oleaginous marine microorganisms, capable of hydrolyzing TG and PC.

Despite the widespread use of lipases in ester synthesis in various environments, including aqueous-organic solvent mixtures and pure organic solvents [7,18], the organic solvent tolerance of lipases from *Fusarium* species has not been extensively studied thus far. FAL enzyme has shown extremely high stability (see Fig 11) at 25% of non-polar organic solvents, after 24 h incubation, with even better stability, 189%-95%, in the presence of cyclohexane, *n*-hexane, n-hexadecane, toluene, chloroform and n-hexanol. The stability of certain lipases in water-immiscible organic solvents is often attributed to their ability to maintain the enzyme hydration layer. This layer is critical for retaining the catalytic activity of the enzyme. By avoiding disruptions to this layer, the lipolytic enzyme can remain stable and active in organic solvents [94]. Moreover, the enzyme did not show a strong decline in residual activity in the presence of certain polar solvents, such as methanol, *n*-butanol, and *iso*-propanol, despite it being well recognized that fungal lipases [48,95] are uncommon stable in miscible-solvents, especially when used at concentrations that would be optimal for the alcoholysis reaction. Our data agrees with Rade et al. [66] who described a novel fungal lipase with high stability toward methanol. Furthermore, Ogino et al. [94,96] and Careri et al. [94,96] have reported a psychrophile solvent tolerant lipase. In fact, in the presence of this water miscible solvent, the hydrophobicity of the medium changes and undergoes a considerable disturbance of the protein hydrophobic core, which leads to protein deactivation. Hence, the remarkable stability of *Fusarium annulatum* lipase in various organic solvents makes it a promising contender for utilization in the ester synthesis process.

The "Green chemicals" are the enzymes used as a substitute for the harmful ingredients in detergents [97]. Most companies are currently producing lipase detergent designed to eliminate the lipid molecules from the soiled substrates [98]. In addition to temperature and pH stability, an exemplary detergent lipase must also be stable in the presence of various detergent ingredients, such as surfactants, bleaching agents, builders, and enzymes [1]. To promote the cleaning process, surfactants are usually used to improve the repulsive force by decreasing the surface tension at the interfaces, while bleaching action occurs via oxidation with the bleaching agents. Anti-redeposition agents are employed to prevent the deposition of stains onto fabrics and to avoid corrosion. To make stain removal more efficient, the catalytic action of the added enzymes is coupled with mechanical action. Recently a new trend in biological laundry detergents consists of the use of an enzyme cocktail (lipases, protease and peroxidase), which has proved to be more efficient.

## Conclusions

This study presents a novel (phospho)lipase family member produced by a newly isolated *Fusarium annulatum* Bugnicourt strain CBS. The enzyme was purified to 62-fold of purity and displays high specific activity on both triglycerides (3500 U/mg on TC8) and on phospholipids, with 5000 U/mg on egg yolk PC. These activities share a similar catalytic triad. When fully characterized, the FAL proved to have a high level of activity in a large pH range and at relatively elevated temperatures, in addition to significant alkaline stability which is infrequent in fungal lipases. In addition, the alkaline lipase has demonstrated remarkable stability when exposed to non-ionic surfactants, oxidizing and bleaching agents, as well as various liquid and

solid detergents. These qualities are particularly noteworthy when compared to the previously studied *Gibbrella zea* lipase GZEL. Our team is currently conducting further research to explore gene cloning and expression, to examine the FAL enzyme structure-function and to construct a 3-D model.

## Supporting information

**S1 File.**
(PDF)

**S2 File.**
(PDF)

## Acknowledgments

This work is dedicated to the memory of Professor T. Doo Hun Kim of the Department of Chemistry at Sookmyung Women's University in Seoul, South Korea. The authors extend their gratitude to B. Kraak and J. Houbraken from the Westerdijk Fungal Biodiversity Institute (WFDI), a part of the Royal Netherlands Academy of Arts and Sciences (KNAW) located in Utrecht, The Netherlands. Their assistance in performing the molecular identification of isolate CBS is greatly appreciated. The authors wish to extend their genuine appreciation to Mr. P. Mansuelle from the Plateforme Protéomique de l'IMM, a part of CNRS and MaP Marseille Protéomique and MaP IBiSA located in Marseille, France. His invaluable assistance in identifying the FAL $NH_2$-terminal is highly acknowledged. Our industrial partners, including Dr. N. Jmal from STE JMAL (EJM)-Laundry Detergent Industry (Boustene, Sfax, Tunisia) and Mr. K. Kacem, Mr. A. Kochtben, and Mr. T. Fdhela from CHIMI-DET, a company of cleaning and detergents products located in El Jem, Mahdia, Tunisia, deserve our heartfelt appreciation. Their technical facility was instrumental in the successful execution of our study, and their generous donation of industrial detergent compounds is greatly acknowledged. The authors' heartfelt acknowledgements are addressed to Dr. I. Boudia (University of M'Sila, Algeria) for his help in analyzing and investigating the contents of the manuscript. English revision by Dr. V. James (Polytech Lyon, Université Lyon 1, France) is also acknowledged.

## Author Contributions

**Conceptualization:** Ahlem Dab, Alexandre Noiriel, Amel Bouanane-Darenfed, Abdelkarim Abousalham, Bassem Jaouadi.

**Data curation:** Ahlem Dab, Ismail Hasnaoui, Sondes Mechri, Alexandre Noiriel.

**Formal analysis:** Ahlem Dab, Ismail Hasnaoui, Sondes Mechri, Ennouamane Saalaoui, Abdeslam Asehraou, Fanghua Wang, Abdelkarim Abousalham, Bassem Jaouadi.

**Funding acquisition:** Abdelkarim Abousalham, Bassem Jaouadi.

**Investigation:** Ahlem Dab, Ismail Hasnaoui, Sondes Mechri, Fawzi Allala, Khelifa Bouacem, Alexandre Noiriel.

**Methodology:** Ismail Hasnaoui, Sondes Mechri, Alexandre Noiriel, Ennouamane Saalaoui, Abdeslam Asehraou, Abdelkarim Abousalham, Bassem Jaouadi.

**Project administration:** Abdelkarim Abousalham, Bassem Jaouadi.

**Resources:** Abdelkarim Abousalham, Bassem Jaouadi.

**Software:** Ismail Hasnaoui, Bassem Jaouadi.

**Supervision:** Abdelkarim Abousalham, Bassem Jaouadi.

**Validation:** Abdelkarim Abousalham, Bassem Jaouadi.

**Visualization:** Sondes Mechri, Fawzi Allala, Khelifa Bouacem, Ennouamane Saalaoui, Abdeslam Asehraou, Fanghua Wang, Abdelkarim Abousalham, Bassem Jaouadi.

**Writing – original draft:** Ahlem Dab, Sondes Mechri, Amel Bouanane-Darenfed, Ennouamane Saalaoui, Abdeslam Asehraou, Fanghua Wang, Abdelkarim Abousalham, Bassem Jaouadi.

**Writing – review & editing:** Alexandre Noiriel, Amel Bouanane-Darenfed, Ennouamane Saalaoui, Abdeslam Asehraou, Fanghua Wang, Abdelkarim Abousalham, Bassem Jaouadi.

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
