## [Decision Letter · Decision Letter 0]

27 Mar 2023

PONE-D-23-02802Biochemical Characterization of an Alkaline and Detergent-Stable Lipase from Fusarium annulatum Bugnicourt Strain CBS Associated with Olive Tree DiebackPLOS ONE

Dear Dr. Jaouadi,

Thank you for submitting your manuscript to PLOS ONE. After careful consideration, we feel that it has merit but does not fully meet PLOS ONE’s publication criteria as it currently stands. Therefore, we invite you to submit a revised version of the manuscript that addresses the points raised during the review process.

Please carefully consider the suggestions done by both reviewers

We look forward to receiving your revised manuscript.

Kind regards,

Guadalupe Virginia Nevárez-Moorillón, Ph.D.

Academic Editor

PLOS ONE

Journal Requirements:

Reviewers' comments:

Reviewer's Responses to Questions

**Comments to the Author**

1. Is the manuscript technically sound, and do the data support the conclusions?

Reviewer #1: Yes

Reviewer #2: Yes

2. Has the statistical analysis been performed appropriately and rigorously? 

Reviewer #1: Yes

Reviewer #2: Yes

3. Have the authors made all data underlying the findings in their manuscript fully available?

Reviewer #1: Yes

Reviewer #2: Yes

4. Is the manuscript presented in an intelligible fashion and written in standard English?

Reviewer #1: Yes

Reviewer #2: Yes

5. Review Comments to the Author

Reviewer #1: Comments

Present study shows the production, purification and characterization of lipase from Fusarium annulatum. Since this study reports that purified enzyme had extreme tolerance to the presence of non-polar organic solvents, non-ionic and anionic surfactants and oxidants, in addition to significant compatibility and stability with some available laundry detergents; the present work is of great significance. This research paper can be accepted for publication in the Journal.

Few queries are as follows:

1. In section “Purification of FAL” my query is whether the mycelium was removed after centrifugation or before centrifugation? Filtration by using Whatmann filter Paper should be done prior to centrifugation.

2. In present study, Dialysis was not performed after ammonium sulfate precipitation. Why? Better results could have been obtained if the dialysis of protein sample was performed.

3. “Fusarium” and scientific names of other organisms should be italicized throughout the manuscript.

4. What is the advantage of isolating the lipase producing fungus from Olive oil trees over other sources? What motivated the authors to go for isolation from this particular source?

5. What was the purpose of extraction of olive oil/triolein in section “Washing Performance Analysis of FAL”?

Reviewer #2: Review comments

MS No: PONE-D-23-02802

The manuscript entitled “Biochemical Characterization of an Alkaline and Detergent-Stable Lipase from Fusarium annulatum Bugnicourt Strain CBS Associated with Olive Tree Dieback” needs major revision. Hence, it is suggested that the authors should address the following comments before the manuscript can be considered for publication.

Major comments

• My major concern is that although the work has been done well, the presented information lacks real novelty. In introduction only – the authors have given a detailed account of commercial lipases which are being utilised in different industrial operations. Moreover, in recent times many papers have been published dealing with lipase characterization from various Fusarium species. Therefore, authors should address the novelty of the present study.

• Lipase screening of all fungal isolates should be shown in comparison to the best one. Kindly provide the images.

• Authors in the text of result section have mentioned twice “Data not shown”. I am concerned how can one believe such claims from authors without backing it by some evidence and/or valid justification.

• The authors conducted a Detergent Compatibility and Stain removal test, though any of the pollution control/waste management applications of lipases would be preferable for improving the standard of this paper.

Minor comments

• Grammatical and spelling errors at several places throughout the text needs to be addressed.

• Kindly check the whether all the figure and table number sequences have been cited in the text.

• (7500 x g)? is this the sign of multiplication × or the alphabet of x ?.

• In Figure 9, spelling of ‘organic solvants’ needs to be rectified to organic solvents.

• The quality of Figure 1 must be improved.

• Check entire text and provide a space before using °C.

6. PLOS authors have the option to publish the peer review history of their article (what does this mean?). If published, this will include your full peer review and any attached files.

Reviewer #1: No

Reviewer #2: No

---

## [Author Response · Author response to Decision Letter 0]

7 May 2023

AUTHORS’ POINT-BY-POINT RESPONSES TO THE JOURNAL REQUIREMENTS AND THE REVIEWER’S COMMENTS

Journal: PLOS ONE

Manuscript ID: PONE-D-23-02802R1

Article Type: Full Length Article

Manuscript Title: Biochemical Characterization of an Alkaline and Detergent-Stable Lipase from Fusarium annulatum Bugnicourt Strain CBS Associated with Olive Tree Dieback in Tunisia: A New Member of the (Phospho)Lipase Family

List of contributing authors: Ahlem Dab, Ismail Hasnaoui, Sondes Mechri, Fawzi Allala, Khelifa Bouacem, Alexandre Noiriel, Amel Bouanane-Darenfed, Ennouamane Saalaoui, Abdeslam Asehraou, Fanghua Wang, Abdelkarim Abousalham, and Bassem Jaouadi

Current status: Revise - Major Revision

Please find below our point-by -point responses to the comments on the manuscript. We are very grateful for the pertinent questions raised and also the suggestions made by the Reviewers. They were valuable in helping us to improve the manuscript’s quality and comprehension. We hope we have clarified and adequately answered all these queries. Detailed changes/corrections according to the journal requirements and Reviewer’s comments are incorporated in the revised manuscript and have been highlighted in RED. Responses to Editor and Reviewer comments are listed in blue, as follows:

JOURNAL REQUIREMENTS:

“When submitting your revision, we need you to address these additional requirements.”

Authors’ Response: Thank you for your helpful comments recommending a number of changes to the structure of our manuscript so as to meet the journal’s submission and GFA standards. We have modified and updated the manuscript accordingly and provided a detailed point-by-point list of corrections below:

1. “Please ensure that your manuscript meets PLOS ONE’s style requirements, including those for file naming. The PLOS ONE style templates can be found at 

https://journals.plos.org/plosone/s/file?id=ba62/PLOSOne_formatting_sample_title_authors_affiliations.pdf.”

Authors’ Response: We have modified it accordingly.

2. “We note that the grant information you provided in the ‘Funding Information’ and ‘Financial Disclosure’ sections do not match. When you resubmit, please ensure that you provide the correct grant numbers for the awards you received for your study in the ‘Funding Information’ section.”

Authors’ Response: We have updated and provided the funding information statement correctly in the online submission system and modified it accordingly for the metadata.

3. “We note that you have stated that you will provide repository information for your data at acceptance. Should your manuscript be accepted for publication, we will hold it until you provide the relevant accession numbers or DOIs necessary to access your data. If you wish to make changes to your Data Availability statement, please describe these changes in your cover letter and we will update your Data Availability statement to reflect the information you provide.”

Authors’ Response: We have modified and updated it accordingly. Thus, please update the text of the “Data Availability Statement” and “Describe where the data may be found in full sentences. If you are copying our sample text, replace any instances of XXX with the appropriate details.” as follows: “Yes – all data are fully available without restriction.” and “All relevant data are within the paper and its Supporting Information files.”, respectively.

4. “PLOS ONE now requires that authors provide the original uncropped and unadjusted images underlying all blot or gel results reported in a submission’s figures or Supporting Information files. This policy and the journal’s other requirements for blot/gel reporting and figure preparation are described in detail at https://journals.plos.org/plosone/s/figures#loc-blot-and-gel-reporting-requirements and https://journals.plos.org/plosone/s/figures#loc-preparing-figures-from-image-files. When you submit your revised manuscript, please ensure that your figures adhere fully to these guidelines and provide the original underlying images for all blot or gel data reported in your submission. See the following link for instructions on providing the original image data: https://journals.plos.org/plosone/s/figures#loc-original-images-for-blots-and-gels. In your cover letter, please note whether your blot/gel image data are in Supporting Information or posted at a public data repository, provide the repository URL if relevant, and provide specific details as to which raw blot/gel images, if any, are not available. Email us at plosone@plos.org if you have any questions.”

Authors’ Response: We have made the necessary changes according to the GFA norms and standards. Additionally, we have modified and updated it accordingly. Thus, please update the text of the “Data Availability Statement” and “Describe where the data may be found in full sentences. If you are copying our sample text, replace any instances of XXX with the appropriate details.” as follows: “Yes – all data are fully available without restriction.” and “All relevant data are within the paper and its Supporting Information files.”, respectively.

5. “We note that you have included the phrase “data not shown” in your manuscript. Unfortunately, this does not meet our data sharing requirements. PLOS does not permit references to inaccessible data. We require that authors provide all relevant data within the paper, Supporting Information files, or in an acceptable, public repository. Please add a citation to support this phrase or upload the data that corresponds with these findings to a stable repository (such as Figshare or Dryad) and provide and URLs, DOIs, or accession numbers that may be used to access these data. Or, if the data are not a core part of the research being presented in your study, we ask that you remove the phrase that refers to these data.”

Authors’ Response: Thank you for your valuable suggestion. We have updated the two-missing data “data not shown” references with new Figures 1 and 3 and modified these phrases accordingly in the revised version of the manuscript.

REVIEWER COMMENTS:

Comments to the Author

Thanks for the constructive comments from the Reviewers. These comments are extremely helpful in improving the readability of our manuscript and in making the paper more solid. The following presents our point-by-point responses as well as the revision of the manuscript.

1. Is the manuscript technically sound, and do the data support the conclusions?

“The manuscript must describe a technically sound piece of scientific research with data that supports the conclusions. Experiments must have been conducted rigorously, with appropriate controls, replication, and sample sizes. The conclusions must be drawn appropriately based on the data presented.”

Reviewer #1: Yes

Reviewer #2: Yes

Authors’ Response: Thank you very much for your appreciation of our work. 

2. Has the statistical analysis been performed appropriately and rigorously?

Reviewer #1: Yes

Reviewer #2: Yes

Authors’ Response: Thank you very much for your appreciation of our work.

3. Have the authors made all data underlying the findings in their manuscript fully available?

“The PLOS Data policy requires authors to make all data underlying the findings described in their manuscript fully available without restriction, with rare exception (please refer to the Data Availability Statement in the manuscript PDF file). The data should be provided as part of the manuscript or its supporting information, or deposited to a public repository. For example, in addition to summary statistics, the data points behind means, medians and variance measures should be available. If there are restrictions on publicly sharing data—e.g. participant privacy or use of data from a third party—those must be specified.”

Reviewer #1: Yes

Reviewer #2: Yes

Authors’ Response: Thank you very much for your favorable response.

4. Is the manuscript presented in an intelligible fashion and written in standard English?

“PLOS ONE does not copyedit accepted manuscripts, so the language in submitted articles must be clear, correct, and unambiguous. Any typographical or grammatical errors should be corrected at revision, so please note any specific errors here.”

Reviewer #1: Yes

Reviewer #2: Yes

Authors’ Response: Thank you very much for your favorable response.

5. Review Comments to the Author

“Please use the space provided to explain your answers to the questions above. You may also include additional comments for the author, including concerns about dual publication, research ethics, or publication ethics. (Please upload your review as an attachment if it exceeds 20,000 characters).”

Authors’ Response: This has been done.

Reviewer #1: Comments

- “Present study shows the production, purification and characterization of lipase from Fusarium annulatum. Since this study reports that purified enzyme had extreme tolerance to the presence of non-polar organic solvents, non-ionic and anionic surfactants and oxidants, in addition to significant compatibility and stability with some available laundry detergents; the present work is of great significance. This research paper can be accepted for publication in the Journal.”

Authors’ Response: The authors would like to thank Reviewer #1 for her/his careful reading and pertinent suggestions which have really helped to improve the current version of the manuscript. We are pleased that the Referee considered our work an important contribution to the field and worth publishing pending a number of minor modifications. We have addressed all general and specific comments made in order to improve the clarity of the manuscript. We agree with all the propositions and have updated our manuscript accordingly. We have also done our best to remove any ambiguity of the data and refined the content. We hope that this revised version will meet the requirements set. In the revised version of the manuscript, we have highlighted in RED all new changes and corrections. We hope that the additions made and introduced in this new version will strengthen the content and status of our study and satisfy the requirements formulated by Reviewer #1. A point-by-point list of responses and explanations is provided below.

The Few queries are as follows:

1. “In section “Purification of FAL” my query is whether the mycelium was removed after centrifugation or before centrifugation? Filtration by using Whatmann filter Paper should be done prior to centrifugation.”

Authors’ Response: We would like to thank the Reviewer for raising this relevant comment and we appreciate the opportunity to clarify this point. In our experiment, we would like to specify that we first removed the mycelium by filtering the culture supernatant using a Whatman filter paper (No. 1) before centrifugation. This initial filtration step helped to separate the majority of the mycelium from the supernatant. Following this, we proceeded with centrifugation at 7500 × g for 20 min to further clarify the supernatant. To ensure complete removal of any remaining mycelium and debris, we performed a second filtration using a Whatman filter paper (No. 1) after centrifugation. In order to take this remark into account, the required information was clarified and provided in the revised version of the manuscript as follows (Page 12, lines 15-21): “The culture medium (500 mL), retained after 5 days of culture of the CBS fungal strain when the lipase activity is at a maximum, was initially filtered through a Whatman grade No. 1 filter paper (110 mm size) to remove the mycelium. The filtrate was then centrifuged for 20 min at 7500 × g to further clarify the supernatant. Following centrifugation, the supernatant was filtered again using a Whatman grade No. 1 filter paper (110 mm size) to ensure complete removal of any remaining mycelium and debris.”

2. “In present study, Dialysis was not performed after ammonium sulfate precipitation. Why? Better results could have been obtained if the dialysis of protein sample was performed.”

Authors’ Response: We would like to clarify that, in this study, we chose not to perform the dialysis immediately after the ammonium sulfate precipitation step for a specific reason. The subsequent purification step involved size exclusion chromatography using a Cytiva Lifescience™ Superdex® 200 Increase 10/300 GL column, which was used to desalt the protein sample and remove any remaining ammonium sulfate. Accordingly, the initial dialysis step was deemed unnecessary. It is essential to note that we performed dialysis later in the purification process. After size exclusion chromatography, the fractions containing lipase activity were pooled, then extensively dialyzed against 20 mM Tris-HCl buffer pH 9 (buffer A). This step effectively removed all interfering substances, including residual ammonium sulfate, before the sample was loaded onto the HiTrap™ Q-Sepharose FF column. In conclusion, although the dialysis was not conducted immediately after the ammonium sulfate precipitation step, it was incorporated at a later stage in the purification process, thus ensuring that the enzyme was appropriately purified and desalted.

3. “Fusarium” and scientific names of other organisms should be italicized throughout the manuscript.

Authors’ Response: Our thanks to the Reviewer for this comment. We apologize for this error in writing the scientific names of microorganisms. As recommended, the entire manuscript (including references) has been revised and the names of the microorganisms are now in italics.

4. What is the advantage of isolating the lipase producing fungus from Olive oil trees over other sources? What motivated the authors to go for isolation from this particular source?

Authors’ Response: Thanks to the Reviewer for this important remark. As recommended, we would like to add the two main reasons for isolating lipase-producing fungi from olive oil trees, specifically in the context of our study. First, Tunisia being one of the world’s largest producers of olive oil, understanding the lipolytic enzyme systems of fungi that contaminate olive trees is of considerable interest. Numerous studies have shown that Fusarium lipase genes are essential for virulence, suggesting that a better understanding of these lipolytic systems could lead to the development of biological pesticides and other solutions to combat fungal contamination in olive oil production. Second, the olive oil industry generates a large amount of oily waste, which could potentially be treated using lipolytic enzymes derived from these fungi. Additionally, in our study, we collected samples from olive trees in the southern region of Tunisia, where the climate is hot and there is a significant amount of oily waste. We hypothesized that fungi able to thrive in such hostile biotopes may produce enzymes that are highly tolerant to extreme conditions. Thus, by isolating lipase-producing fungi from this particular source, our aim was to identify enzymes with unique properties that could be applied to several biotechnological applications, such as oily waste treatment, thus contributing to more sustainable olive oil production practices.

5. What was the purpose of extraction of olive oil/triolein in section “Washing Performance Analysis of FAL”?

Authors’ Response: The authors gratefully acknowledge Reviewer #1 for raising this point. As requested, we would like to add that, in the required sub-section, the purpose of extracting olive oil/triolein was to evaluate the efficiency of FAL, GZEL, or Lipolase® in removing oily stains when used with a commercial detergent. In fact, we tested the washing performance of these enzymes on cotton cloth stained with tomato sauce, ketchup, or egg yolk. After washing the stained fabric with various treatments using the enzymes and the detergent, we assessed the stain removal efficiency. The extraction of olive oil/triolein allowed us to quantify the amount of oil removed from the fabric and calculate the percentage of oil removal. This helped us determine the effectiveness of FAL, GZEL, or Lipolase® as bio-detergent additives.

Reviewer #2: Review Comments

“The manuscript entitled “Biochemical Characterization of an Alkaline and Detergent-Stable Lipase from Fusarium annulatum Bugnicourt Strain CBS Associated with Olive Tree Dieback” needs major revision. Hence, it is suggested that the authors should address the following comments before the manuscript can be considered for publication.”

Authors’ Response: The authors would like to thank Reviewer #2 for her/his careful reading and pertinent suggestions which really helped to improve the current version of the manuscript. We have addressed all general and specific comments made in order to clarify the manuscript. We agree with all the propositions which were valuable in helping us improve the manuscript’s quality and comprehension. In the revised version of the manuscript, we have highlighted all changes and corrections. A point-by-point list of responses and explanations is provided below.

Major comments

● “My major concern is that although the work has been done well, the presented information lacks real novelty. In introduction only – the authors have given a detailed account of commercial lipases which are being utilised in different industrial operations. Moreover, in recent times many papers have been published dealing with lipase characterization from various Fusarium species. Therefore, authors should address the novelty of the present study.”

Authors’ Response: We appreciate your concern regarding the novelty of our study. While it is true that numerous studies have been published on the characterization of lipases from various Fusarium species, only a handful of them have actually explored the application of these enzymes in detergents, as you mentioned. Our study offers several novel aspects that distinguish it from previous research. First, our study provides an in-depth characterization of not only the Fusarium annulatum lipase (named FAL) but also the Fusarium graminearum lipase (named GZEL). This comprehensive analysis allows for a better understanding of their potential applications in the detergent industry. Second, and perhaps most importantly, our newly characterized enzyme demonstrates the ability to hydrolyze not only triglycerides but also phospholipids, which is a highly desirable characteristic for detergent applications. We have shown, for the first time, that FAL activity on phospholipids is of the PLA1 type, using chemically defined phospholipids synthesized in the laboratory. The broad substrate range of our enzyme enables it to efficiently eliminate various lipid stains, making it an ideal candidate for detergent formulations. Another novel aspect lies in the fact that this is the first time Fusarium annulatum has been identified in Tunisian olive trees associated with Olive Tree Dieback. Numerous studies have highlighted the correlation between the lipolytic enzyme system of Fusarium species and certain plant dieback phenomena. Thus, our work not only contributes to the development of novel enzymatic solutions for the detergent industry but also sheds light on the potential role of Fusarium annulatum in Olive Tree Dieback, which could have implications for agricultural management strategies.

● “Lipase screening of all fungal isolates should be shown in comparison to the best one. Kindly provide the images.”

Authors’ Response: We appreciate the Reviewer’s suggestion to provide a visual comparison of lipase screening for all fungal isolates relative to the best-performing isolate. To address this concern, we have added supplementary images that display the lipase screening results for all fungal isolates (10), with a clear indication of the best-performing CBS fungal isolate. These images illustrate the lipolytic activity of each isolate on the PDA plates, enabling readers to visually compare their performances.

To take this comment into account, we have specified that “Ten lipase-producing fungi (TN10, M45, AS15, AI16, R22, CBS, F6, P63, C58, and V35) were obtained showing lipase activity after 5 days of incubation at 25°C (Fig 1).” and a new figure (named Fig. 1) has been added in the revised version of the manuscript (Page 20, lines 12-14).

● “Authors in the text of result section have mentioned twice “Data not shown”. I am concerned how can one believe such claims from authors without backing it by some evidence and/or valid justification.”

Authors’ Response: We would like to thank the Reviewer for making this pertinent comment. As requested, in order to take this point into account, we would like to state that the two “data not shown” have now been provided, as Figure 1 and Figure 3, in the Results section of the revised version of the manuscript, as follows:

- “Ten lipase-producing fungi (TN10, M45, AS15, AI16, R22, CBS, F6, P63, C58, and V35) were obtained showing lipase activity after 5 days of incubation at 25 °C (Fig 1).” (Page 20, lines 12-14).

- “The data revealed that the fungal strain CBS culture had a characteristic deep purple pigmentation on PDA (Fig 3).” (Page 21, lines 3-4).

Additionally, new figures are cited in the text and added in the revised version of the manuscript in the “Figure Legends” section as follows:

Figure 1. Fluorescent haloes of lipase-producing fungi, on rhodamine B with 1% (v/v) olive oil agar medium, were visible under UV light at 365 nm. Ten lipase-producing fungi (TN10, M45, AS15, AI16, R22, CBS, F6, P63, C58, and V35) were obtained showing variable lipase activities. After incubation, for 5 days at 25 °C, plates were subjected to UV irradiation and photographed. (Page 53, lines 2-6).

Figure 3. Colony of strain CBS isolated from soil-borne fungi of the olive tree, Olea europaea cv. Chemlali. (A) Upper view of a colony on PDA. (B) reverse view of colony on PDA. (Page 53, lines 16-18).

● “The authors conducted a Detergent Compatibility and Stain removal test, though any of the pollution control/waste management applications of lipases would be preferable for improving the standard of this paper.”

Authors’ Response: We appreciate the Reviewer’s suggestion to explore pollution control and waste management applications of lipases in our study. While the current paper focuses primarily on detergent compatibility and stain removal testing, we recognize the importance and potential of these enzymes in environmental applications. Therefore, we are planning to investigate the use of Fusarium annulatum lipase (FAL) in pollution control and waste management in our upcoming research projects. In our future studies, we intend to evaluate the ability of these lipases to degrade lipids in wastewater samples, assess their efficiency in removing oil and grease from various sources, and explore their potential role in the bioremediation of contaminated soils. Additionally, we will investigate the impact of these lipases on waste management, specifically in the context of converting lipid waste into valuable products such as biodiesel or other bio-based materials. By including these additional applications in our upcoming research, we aim to emphasize the versatility of the lipases in question and underscore their potential contribution to sustainable and environmentally friendly solutions in various industries.

Minor comments

● “Grammatical and spelling errors at several places throughout the text needs to be addressed.”

Authors’ Response: Thanks to the Reviewer for pointing this out. As recommended, we have subjected the manuscript to language editing by submitting it for proofreading and language polishing to a native English-speaking researcher, Associate Professor Dr. Valérie JAMES from the Polytech Lyon (Université Lyon 1, France). We hope we have clarified and adequately answered all these queries. We believe that in its current form, the language of our paper meets the standards required by Reviewer #2 and the PLOS ONE and the revised form of this manuscript has taken the shape of a conventional paper.

● “Kindly check the whether all the figure and table number sequences have been cited in the text.”

Authors’ Response: As requested, all figures and tables are now labeled with their names, structured and correctly checked, and cited in sequence in the text.

● “(7500 x g)? is this the sign of multiplication × or the alphabet of x?”

Authors’ Response: Apologies for the confusion caused by the use of letter "x" in the expression "7500 x g." In this context, the "x" represents the multiplication symbol, and the correct expression should be "7500 × g." This expression denotes a centrifugation force of 7500 times the acceleration due to gravity (g), which is a common notation used in scientific literature when describing centrifugation steps in experimental procedures.

● “In Figure 9, spelling of ‘organic solvants’ needs to be rectified to organic solvents.”

Authors’ Response: We apologize for this mistake. As recommended ‘organic solvants’ has been replaced by ‘organic solvents’ in Figure 9 (now Figure 11 in the revised version of the manuscript).

● “The quality of Figure 1 must be improved.”

Authors’ Response: On the Reviewer’s good advice, we have tried our best to improve the quality of Figure 1 (now Figure 4 in the revised version of the manuscript) and the other figures. A 1200 DPI higher resolution, instead of 300 DPI, for printing all the figures was generated by the GIMP 2.10.20 program (https://www.gimp.org/) and is provided in the revised manuscript. We believe that the quality of the figures in their current form is simpler and clearer and will meet the standards required by Reviewer #2 and will comply with the GFA of the journal.

● “Check entire text and provide a space before using °C.”

Authors’ Response: As per the Reviewer’s advice, we have ensured that a space has been provided before each instance of "°C".

All these modifications have been introduced to the revised manuscript as per the editor and reviewers’ instructions. We hope that this revised version will meet the standards required by the Editor and Reviewers and will be favorably considered for publication in the prestigious journal, “PLOS ONE”.

---

## [Editor Report · Decision Letter 1]

9 May 2023

Biochemical Characterization of an Alkaline and Detergent-Stable Lipase from Fusarium annulatum Bugnicourt Strain CBS Associated with Olive Tree Dieback

PONE-D-23-02802R1

Dear Dr. Jaouadi,

We’re pleased to inform you that your manuscript has been judged scientifically suitable for publication and will be formally accepted for publication once it meets all outstanding technical requirements.

Kind regards,

Guadalupe Virginia Nevárez-Moorillón, Ph.D.

Academic Editor

PLOS ONE
---

## [Editor Report · Acceptance letter]

11 May 2023

PONE-D-23-02802R1 

Biochemical Characterization of an Alkaline and Detergent-Stable Lipase from *Fusarium annulatum* Bugnicourt Strain CBS Associated with Olive Tree Dieback 

Dear Dr. Jaouadi:

I'm pleased to inform you that your manuscript has been deemed suitable for publication in PLOS ONE. Congratulations! Your manuscript is now with our production department. 

Kind regards, 

on behalf of

Dr. Guadalupe Virginia Nevárez-Moorillón 

Academic Editor

PLOS ONE